# EcR recruits dMi-2 and increases efficiency of dMi-2-mediated remodelling to constrain transcription of hormone-regulated genes

Judith Kreher[1],[*],[†], Kristina Kovač[1],[*], Karim Bouazoune[1], Igor Mačinković[1], Anna Luise Ernst[1],[†], Erik Engelen[1],[†], Roman Pahl[2],[†], Florian Finkernagel[3], Magdalena Murawska[1],[†], Ikram Ullah[1] & Alexander Brehm[1]

Gene regulation by steroid hormones plays important roles in health and disease. In *Drosophila*, the hormone ecdysone governs transitions between key developmental stages. Ecdysone-regulated genes are bound by a heterodimer of ecdysone receptor (EcR) and Ultraspiracle. According to the bimodal switch model, steroid hormone receptors recruit corepressors in the absence of hormone and coactivators in its presence. Here we show that the nucleosome remodeller dMi-2 is recruited to ecdysone-regulated genes to limit transcription. Contrary to the prevalent model, recruitment of the dMi-2 corepressor increases upon hormone addition to constrain gene activation through chromatin remodelling. Furthermore, EcR and dMi-2 form a complex that is devoid of Ultraspiracle. Unexpectedly, EcR contacts the dMi-2 ATPase domain and increases the efficiency of dMi-2-mediated nucleosome remodelling. This study identifies a non-canonical EcR-corepressor complex with the potential for a direct regulation of ATP-dependent nucleosome remodelling by a nuclear hormone receptor.

[1] Institute of Molecular Biology and Tumour Research, Philipps University Marburg, Marburg 35037, Germany. [2] Institute of Medical Biometry and Epidemiology, Philipps University Marburg, Marburg 35037, Germany. [3] Center for Tumour Biology and Immunology, Philipps University Marburg, Marburg 35043, Germany. * These authors contributed equally to this work. † Present addresses: Roche Diagnostics Deutschland GmbH, Mannheim 68305, Germany (J.K. and E.E.); Institute of Molecular Biology, Mainz 55128, Germany (A.L.E.); GSK Marburg Vaccines and Diagnostics, Marburg 35041, Germany (R.P.); Biomedical Center Munich, Ludwig Maximilians University, Planegg-Martinsried 82152, Germany (M.M.). Correspondence and requests for materials should be addressed to A.B. (email: brehm@imt.uni-marburg.de).

In *Drosophila*, the hormone 20-hydroxy ecdysone (20HE) controls major developmental transitions[1]. Ecdysone binds to a heterodimeric nuclear hormone receptor composed of the ecdysone receptor (EcR) and the *Drosophila* RXR homolog Ultraspiracle (USP). In the absence of hormone EcR–USP binds DNA and represses transcription by interacting with corepressors[2–4]. Hormone exposure increases transport of EcR and USP into the nucleus and conformational changes result in the exchange of corepressors for coactivators and gene activation[5,6]. In this bimodal switch model, EcR–USP serves as a relatively static landing platform for several corepressors and coactivators that can modify histones and remodel chromatin. The EcR–USP heterodimer shares this mechanism with mammalian class II nuclear hormone receptors. However, recent results suggest that gene regulation by nuclear hormone receptors is more complex than this model implies: nuclear receptor complex formation and their binding to chromatin as well as to coregulators are highly dynamic[7,8].

Vertebrate CHD3 (Mi-2-alpha) and CHD4 (Mi-2-beta) and *Drosophila* Mi-2 (dMi-2) are members of the CHD family of ATP-dependent nucleosome remodellers. They are central subunits of Nucleosome Remodelling and Deacetylation (NuRD) complexes and play important roles in development[9–12]. NuRD and CHD4 are indispensable for proper blastocyst and embryonic stem cell differentiation[13,14]. Moreover, CHD4 and dMi-2 are important for cell fate determination in several developmental lineages where they cooperate with transcription factors to establish differentiation-specific transcription programmes by generating chromatin environments conducive to gene repression or activation[15–17].

Here we demonstrate accumulation of dMi-2 at ecdysone-activated polytene chromosome puffs by immunofluorescence. We used chromatin immunoprecipitation sequencing (ChIP-seq) to identify genomic regions to which dMi-2 binds in response to ecdysone treatment of S2 cells. A high number of these regions map to classical ecdysone target genes, such as *vrille* and *broad complex* (*Br-C*). RNA interference and quantitative reverse transcription PCR (RT–qPCR) analyses revealed that dMi-2 limits the transcription of coding and noncoding RNAs emanating from ecdysone-regulated genes thereby restricting the dynamic range of their activation. In agreement with a repressive role, dMi-2 is required for maintaining a closed chromatin conformation at the *vrille* locus, as detected by micrococcal nuclease (MNase) digestion. Knockdown and ChIP experiments show that dMi-2 recruitment depends on the EcR subunit but, surprisingly, not on the USP subunit of the heterodimer. Biochemical analysis identified the formation of a EcR–dMi-2 complex that is devoid of USP. dMi-2 and USP interact with the same domain of EcR and bind in a mutually exclusive manner. Unexpectedly, EcR directly contacts the ATPase domain of dMi-2 and increases the efficiency of dMi-2-mediated nucleosome remodelling *in vitro*. Our results reveal a non-canonical nuclear hormone receptor–corepressor complex and a novel relationship between hormone receptor and nucleosome remodelling.

## Results

**dMi-2 associates with the ecdysone-activated Br-C locus.** As larvae progress through development, several genes are activated by the hormone 20HE[1]. Gene activation results in dramatic changes in chromatin structure that visually manifest themselves as polytene chromosome 'puffs'. We observed a marked enrichment of dMi-2 at prototypical early ecdysone-induced puffs, including bands B2 (containing the *broad complex* (*Br-C*) locus), 74EF and 75B (Fig. 1a). The ISWI ATPase was

excluded from these puffs demonstrating that the observed accumulation of dMi-2 was specific for this remodeller. *Br-C* is an early ecdysone target and encodes several zinc finger transcription factors that activate genes at later stages of the ecdysone cascade (Supplementary Fig. 1A). In S2 cells, ecdysone exposure strongly activates *Br-C* transcription without affecting dMi-2 expression (Supplementary Fig. 1B,C). Published dMi-2 ChIP-chip and ChIP-seq data suggested robust dMi-2 binding within the first intron of the major *Br-C* transcripts (modENCODE data sets Q.2626.S2 and Q4443.S2;[18]). We verified dMi-2 association with this region by ChIP (Supplementary Fig. 1D). Moreover, we detected increased dMi-2 binding to *Br-C* when cells were exposed to ecdysone for 6 h. These observations demonstrate that dMi-2 associates with the ecdysone-regulated *Br-C* locus in larvae and S2 cells and that the strength of this association can be modulated by hormone. This raised the question whether dMi-2 also binds and regulates other ecdysone-activated genes.

**Ecdysone increases dMi-2 chromatin binding.** We performed ChIP-seq to identify genomic regions displaying an ecdysone-induced increase in dMi-2 association in S2 cells. Comparison of ChIP-seq profiles in the absence and presence of ecdysone identified 185 such regions (tag count ratio treated versus untreated of ≥ 2.3; Supplementary Data 1). From here on, we refer to these regions as ecdysone-induced dMi-2 binding regions (EIMRs). EIMRs strongly correlated with well-established ecdysone-induced genes (36% of the top 25, 24% of the top 50 EIMRs), including *Br-C*, *vrille*, *Ecdysone-induced protein* (*Eip*) genes *Eip74EF* and *Eip75B* and *let-7* (Fig. 1b). We verified that these genes were activated by ecdysone using RT–qPCR (Fig. 1c). We next inspected the dMi-2 ChIP-seq profiles of two well-established early ecdysone targets, *Br-C* and *vrille*, in detail. Several EIMRs mapped within a region surrounding the transcriptional start site of the *Br-C* transcripts *broad-RA* and *-RB* (Fig. 1b,d). Notably, this region contained clear dMi-2 ChIP-seq signals even in the absence of ecdysone. The *vrille* locus harboured two regions with prominent EIMRs (Fig. 1e). Again, these regions bound dMi-2 also in the absence of hormone. We validated hormone-modulated dMi-2 association with *Br-C* and *vrille* and the specificity of the ChIP-seq results by RNAi and ChIP-qPCR (Supplementary Fig. 1e–g). Taken together, the ChIP analyses suggest that ecdysone treatment does not generate *de novo* dMi-2-binding sites. Rather, the hormone increases the level of dMi-2 chromatin association at specific regions within ecdysone-regulated genes.

**dMi-2 fine-tunes the kinetics and constrains gene activation.** We next asked whether dMi-2 was regulating transcription of *Br-C* and *vrille*. We depleted EcR, its heterodimerization partner USP or dMi-2 by RNAi in S2 cells (Fig. 2a,b), exposed cells to ecdysone and then followed *Br-C* and *vrille* transcript levels over the course of 6 h by RT–qPCR (Fig. 2c,d). In control cells, *Br-C* and *vrille* were efficiently stimulated by ecdysone. As expected, depletion of EcR abrogated activation. Unexpectedly, depletion of USP still allowed robust stimulation of both genes. Depletion of dMi-2 markedly increased ecdysone-mediated activation of both genes. By contrast, RNAi-mediated depletion of ISWI, an unrelated chromatin remodeller, did not significantly affect *Br-C* and *vrille* activation in this system (Supplementary Fig. 2). We conclude that EcR is essential for hormone-mediated stimulation of *Br-C* and *vrille*, whereas its dimerization partner USP is largely dispensable. dMi-2 appears to fine-tune stimulation kinetics and prevents excessive *Br-C* and *vrille* activation.

**dMi-2 represses ecdysone-inducible genes.** Given that the EcR–USP heterodimer represses transcription in the absence of

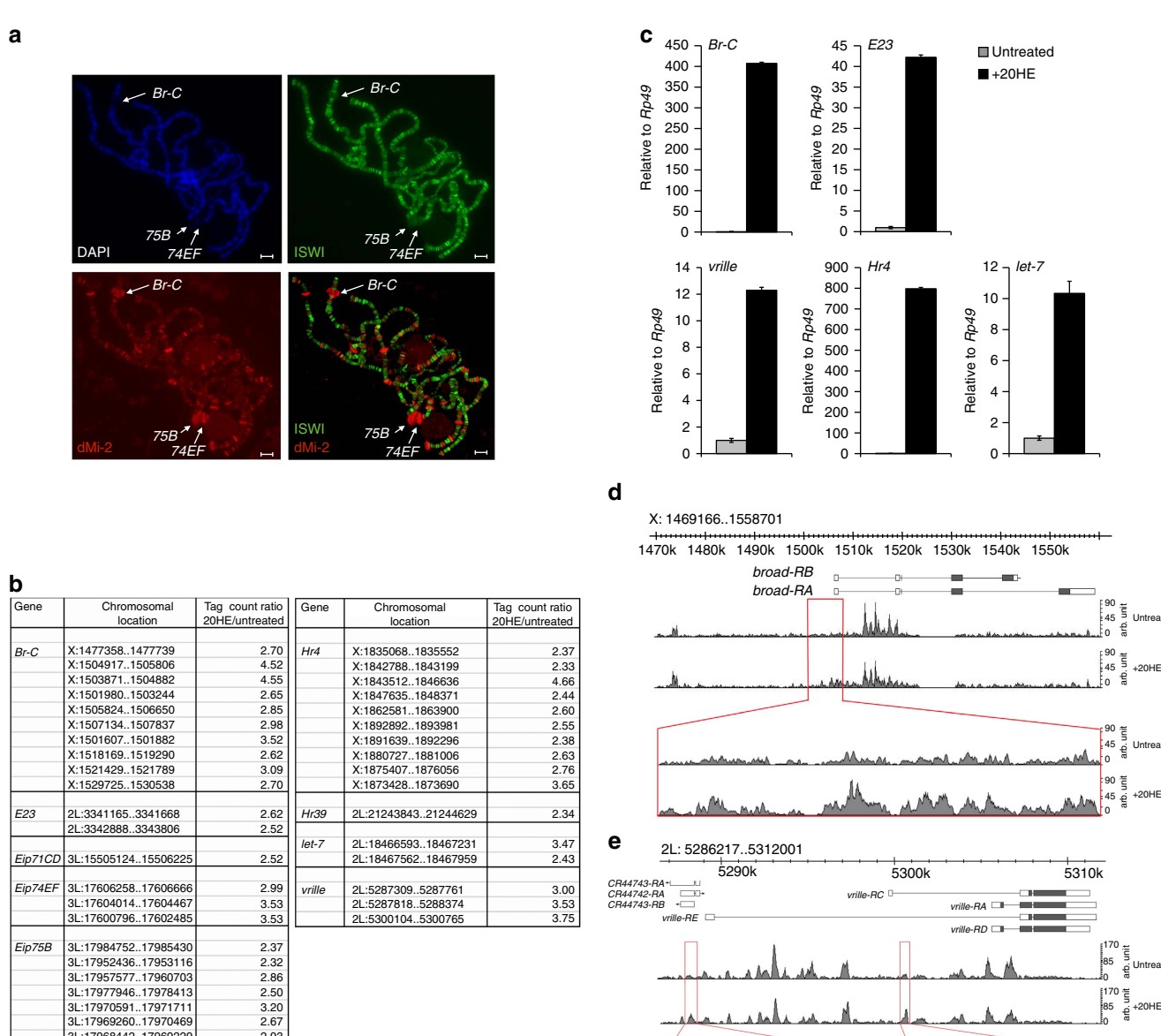

**Figure 1 | dMi-2 binds ecdysone-activated genes.** (**a**) Polytene chromosomes were stained with DAPI (upper left panel), ISWI antibody (upper right panel) or dMi-2 antibody (lower left panel) and analysed by immunofluorescence. Lower right panel shows overlay of ISWI and dMi-2 signals. Arrows indicate three early ecdysone-induced puffs: *Br-C*, *74EF*, and *75B*. Scale bar is 10 μm. (**b**) Regions with ecdysone-induced increased dMi-2 binding (EIMRs) within known ecdysone-activated genes. Cells were treated for 6 h with 1 μM 20HE. (**c**) RNA expression of EIMR-containing genes in S2 cells was determined by RT–qPCR. Ratios of RNA levels of the gene of interest to *rp49* RNA levels in untreated and 20HE-treated cells were calculated. The ratio determined in untreated cells was set to 1 and the ratio in ecdysone-treated cells was expressed relative to this. Error bars denote s.d. of technical triplicates. (**d**) dMi-2 ChIP-seq profile across the *Br-C* locus in untreated and ecdysone-treated (+20HE) S2 cells. Top: schematic representation of *Br-C* locus. Only two *Br-C* transcripts are shown for clarity; see Supplementary Fig. 1a for complete set. Bottom: magnification of most prominent EIMR-containing region within *Br-C*. (**e**) dMi-2 ChIP-seq profile across the *vrille* locus in untreated and ecdysone-treated (+20HE) S2 cells. Top: schematic representation of *vrille* locus. Bottom: magnification of two EIMR-containing regions.

hormone[2–4], we also analysed the transcript levels of ecdysone-regulated genes in RNAi-treated cells before ecdysone was added (Fig. 2e). In addition to *Br-C* and *vrille*, we also included in the analysis *Hr4*, *E23*, *let-7* and two overlapping noncoding RNAs that are transcribed from different strands within an EIMR-containing region upstream of the *vrille RE* promoter (*CR44743* and *CR44742*; Fig. 1e). As expected, depletion of EcR resulted in the derepression of the majority of genes tested (1.5-fold to 20-fold). By

contrast, with the exceptions of *Br-C* and *E23* (2-fold increase in transcription), USP depletion failed to significantly derepress these genes. When we depleted dMi-2, all genes analysed were robustly upregulated. Derepression levels ranged from 2.5-fold (*vrille* and *Hr4*) to 50-fold (*let-7*). Again, depletion of the ISWI chromatin remodeller did not produce such effects (Supplementary Fig. 2, time point 0′). These results suggest that both EcR and dMi-2 play important roles in repressing the basal transcription of ecdysone-

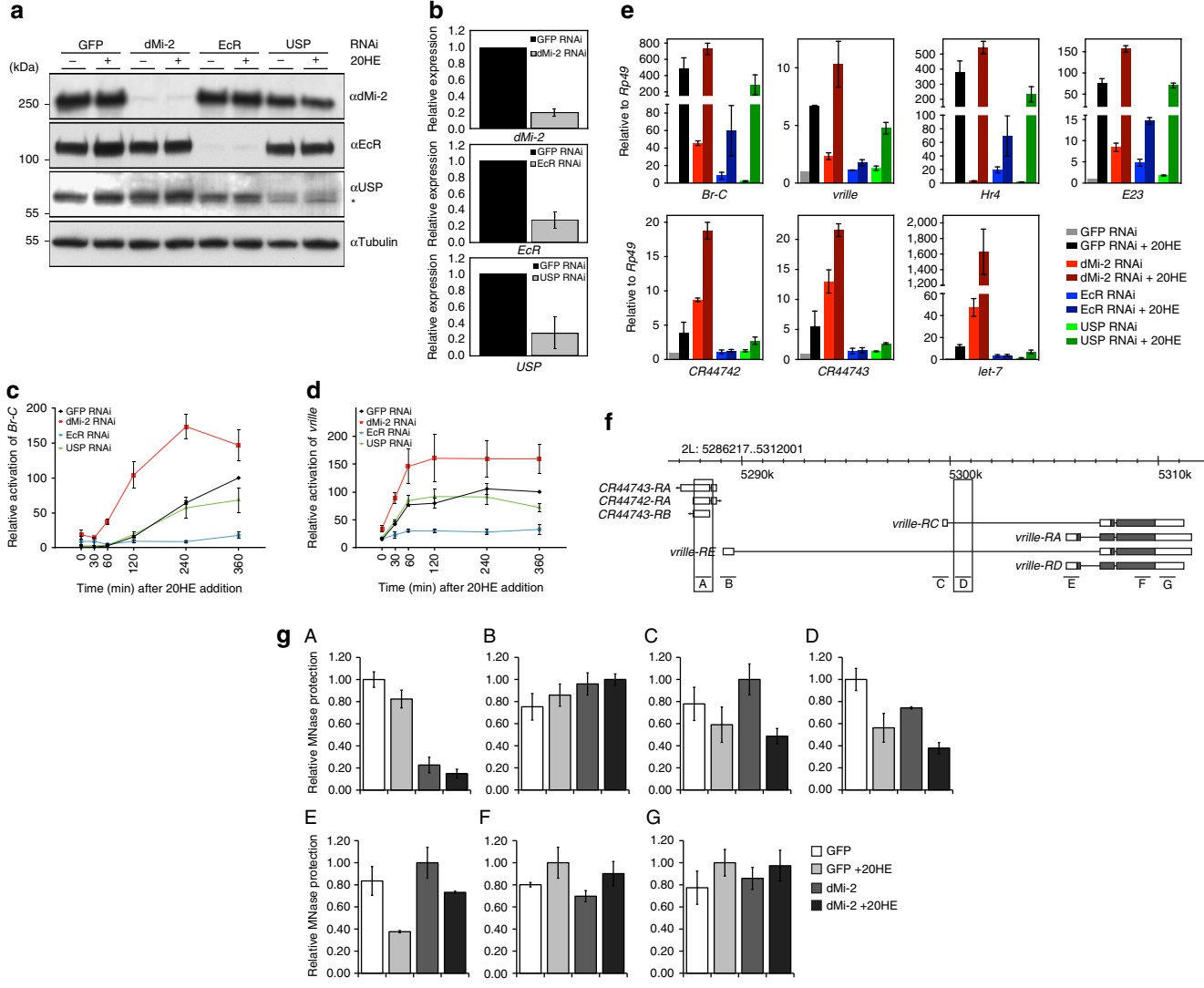

**Figure 2 | dMi-2 constrains the transcription of ecdysone-activated genes and contributes to a closed chromatin structure. (a)** S2 cells were RNAi-depleted of GFP (control), dMi-2, EcR or USP and either left untreated ( − 20HE) or were treated with 1 µM ecdysone for 6 h ( + 20HE). Nuclear extracts were prepared and analysed by western blot. Asterisk (\*) denotes USP antibody-crossreactive protein. Tubulin served as loading control. Representative examples of three independent RNAi experiments are shown. **(b)** RT–qPCR analysis of dMi-2 (top), EcR (middle) and USP (bottom) expression. RNA levels in control cells (GFP RNAi) were set to 1 and RNA levels in other RNAi-treated cells are depicted relative to the level in control cells. **(c)** Time course of Br-C and **(d)** vrille RNA expression over 6 h of ecdysone treatment. Before ecdysone addition, cells were RNAi-depleted of GFP (control), dMi-2, EcR or USP as indicated. RNA levels were determined by RT–qPCR relative to rp49 and adjusted to 100 at the latest time point of control cells (GFP RNAi). Other ratios were expressed relative to this. Data shown are mean value ± s.e.m. of three independent experiments. **(e)** Derepression of ecdysone-regulated genes after dMi-2, EcR and USP depletion. S2 cells were RNAi-depleted of GFP (control), dMi-2, EcR or USP. Cells were then left untreated or were treated for 6 h with 1 µM 20HE as shown. Data shown are mean value ± s.e.m. of two independent experiments. **(f)** Schematic representation of amplimer positions (A–G) within the vrille locus. Note that regions A and D contain EIMRs (see Fig. 1e). **(g)** Cells were RNAi-depleted of GFP (control) or dMi-2 and then left untreated or treated with ecdysone ( + 20HE). Chromatin was prepared, digested with MNase and amplified by qPCR. Relative MNase protection was calculated and plotted (see Methods section). The sample with the highest MNase protection value for each genomic region was set to 1. Error bars denote s.d. of technical triplicates. Experiment was performed as biological triplicates, one representative experiment is shown.

dependent genes in the absence of hormone. Moreover, our results indicate that repression by EcR and dMi-2 can be maintained even when USP levels are greatly reduced.

When we analysed transcription of the same set of genes after 6 h exposure to 20HE, we observed strong reduction of gene activation in EcR-depleted cells (ranging from 1.5-fold to 70-fold reduction) but not in USP-depleted cells. Depletion of dMi-2 resulted in a general increase in gene activation.

In summary, our depletion experiments reveal that dMi-2 represses basal transcription and limits ecdysone-induced activation of several coding and noncoding transcripts.

**dMi-2 contributes to a closed chromatin structure.** We asked whether dMi-2 regulates transcription by modulating chromatin structure. To address this question, we used MNase digestion of chromatin coupled to qPCR to assess changes in chromatin accessibility at the vrille locus following RNAi depletion of dMi-2 (Fig. 2f,g). Regions with a more accessible chromatin structure are more sensitive to MNase digestion resulting in lower qPCR product levels. In the absence of ecdysone, most vrille regions interrogated did not show significant changes in chromatin accessibility following dMi-2 depletion. However, both EIMR-containing

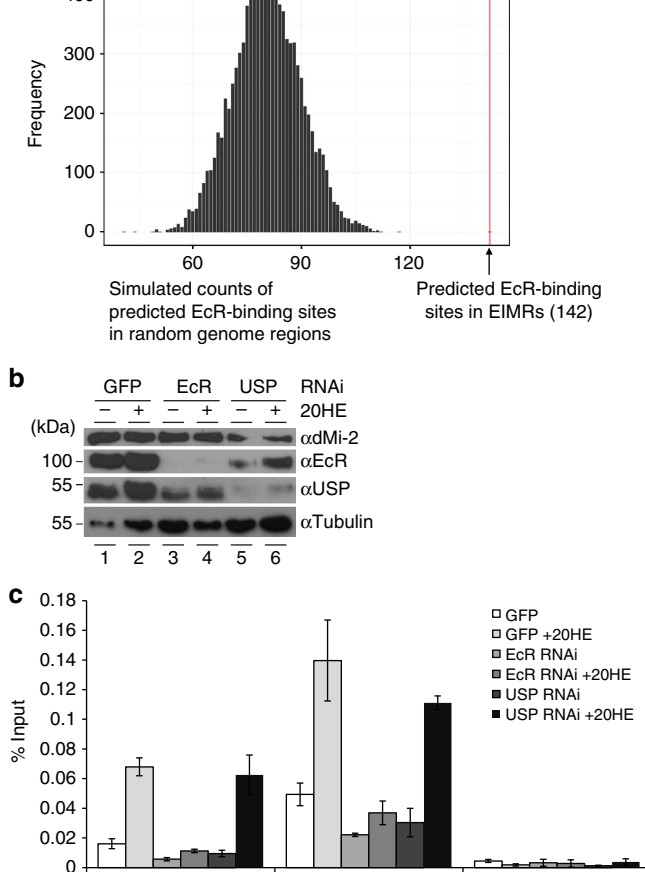

**Figure 3 | EcR but not USP is essential for recruitment of dMi-2 to ecdysone-activated genes.** (**a**) Enrichment of predicted EcR–USP-binding sites in EIMRs. The histogram shows the distribution (sampled from 10,000 runs) of putative EcR–USP site number within the same number of randomly selected genomic regions with the same sizes as the experimentally determined EIMRs. The red line indicates the number of computationally identified EcR–USP-binding motifs within EIMRs (142; $P < 10\text{-}8$). The $P$-value for the observed number of binding motifs was estimated from the normal distribution that can be approximated based on the displayed simulated distribution, that is, using mean and s.d. estimates from the simulated sample. (**b**) Western blot analysis of protein extracts of S2 cells treated with dsRNA directed against GFP (control), EcR or USP. Cells were either left untreated ($-20$HE) or were treated with ecdysone ($+20$HE). Antibodies used are shown on the right, molecular masses on the left. Tubulin served as a loading control. (**c**) dMi-2 binding to *Br-C* and *vrille* following EcR and USP depletion analysed by ChIP-qPCR. Chromatin was prepared from S2 cells that were first treated with dsRNA against GFP (control), EcR or USP and then either left untreated or were exposed to 1 μM ecdysone for 6 h as indicated. ChIP-qPCR was performed with dMi-2 antibody. Error bars denote s.d. of technical triplicates. Experiments were performed as biological triplicates. One representative example is shown.

*vrille* regions displayed increased MNase sensitivity (region A: 80% reduction of qPCR signal, region D: 20% reduction of qPCR signal). We observed similar results in ecdysone-exposed cells.

Taken together, these results suggest that dMi-2 maintains a more closed, inaccessible chromatin structure at EIMRs both in the absence and presence of ecdysone. The loss of MNase

protection in dMi-2 depleted cells correlates with derepression of basal and excessive activation of ecdysone-induced *vrille* transcription (Fig. 2b,c). These findings support the hypothesis that dMi-2 represses ecdysone-induced genes, at least in part, by generating closed, less accessible chromatin structures.

**EcR but not USP is required for dMi-2 recruitment**. We analysed EIMR DNA sequences bioinformatically and found a strong enrichment of predicted EcR–USP-binding sites (Fig. 3a). This suggests that EcR–USP plays a role in dMi-2 recruitment. To test this, we RNAi-depleted S2 cells of EcR or USP (Fig. 3b), exposed cells to ecdysone and determined dMi-2 binding to *Br-C* and *vrille* by ChIP-qPCR (Fig. 3c). The increase of dMi-2 binding to *Br-C* and *vrille* after ecdysone exposure was abolished in EcR-depleted cells. Surprisingly, depletion of USP had only a minor effect on dMi-2 chromatin binding. These results reveal that efficient recruitment of dMi-2 to *Br-C* and *vrille* is critically dependent on EcR but not significantly affected by USP depletion.

**EcR and dMi-2 interact *in vivo* and *in vitro*.** To determine whether dMi-2, EcR and USP physically interact, we immunoprecipitated dMi-2 from S2 nuclear extracts from untreated (Fig. 4a, lanes 1–4) or ecdysone-exposed cells (lanes 5–8). As expected, dMi-2 antibody but not control antibodies precipitated dMi-2 (lanes 2–4 and 6–8). EcR was clearly detectable in both anti-dMi-2 immunoprecipitates. Interestingly, this interaction was not significantly influenced by hormone (compare lanes 4 and 8), suggesting that dMi-2 and EcR form a complex in an ecdysone-independent manner *in vivo*. Given that the USP western blot signals were generally weak (compare dMi-2, EcR and USP signals in lanes 1 and 5) our failure to coprecipitate USP with dMi-2 did not allow us to rule out a physical interaction in this experiment.

We confirmed the interaction of EcR and dMi-2 using baculovirus-expressed recombinant proteins (Fig. 4b and Supplementary Fig. 3A). Both proteins interacted strongly, as judged by Coomassie staining, and the complex was resistant to high salt concentrations (Fig. 4b). Treating baculovirus-infected Sf9 cells with ecdysone did not influence binding (Supplementary Fig. 3B). We also pretreated the serum used for culturing cells with charcoal to deplete any steroid hormone traces that might influence dMi-2 binding to EcR. This did not affect the result. Taken together, these findings suggest that EcR and dMi-2 interact in an ecdysone-independent manner both *in vitro* and *in vivo*.

**dMi-2 and USP compete for binding to EcR**. We used baculovirus-expressed recombinant proteins to analyse the interactions between dMi-2, EcR and USP in more detail (Fig. 4c, top left panel). We co-expressed FLAG-dMi-2, untagged EcR and HA–USP in different combinations and subjected extracts to FLAG- or HA-affinity purification. Again, dMi-2 copurified with EcR (middle left panel). By contrast, USP was undetectable in the dMi-2 immunoprecipitate. Purification of USP demonstrated a robust interaction with EcR (bottom left panel). By contrast, no dMi-2 was detectable in the USP immunoprecipitate. Thus we were unable to detect an interaction between USP and dMi-2 even when both proteins were strongly overexpressed.

Since both dMi-2 and USP bound to EcR, we asked whether they could do so simultaneously. When we co-expressed all three proteins, EcR still efficiently bound to USP (Fig. 4c, bottom left panel, lane 40). By contrast, EcR was barely detectable in the

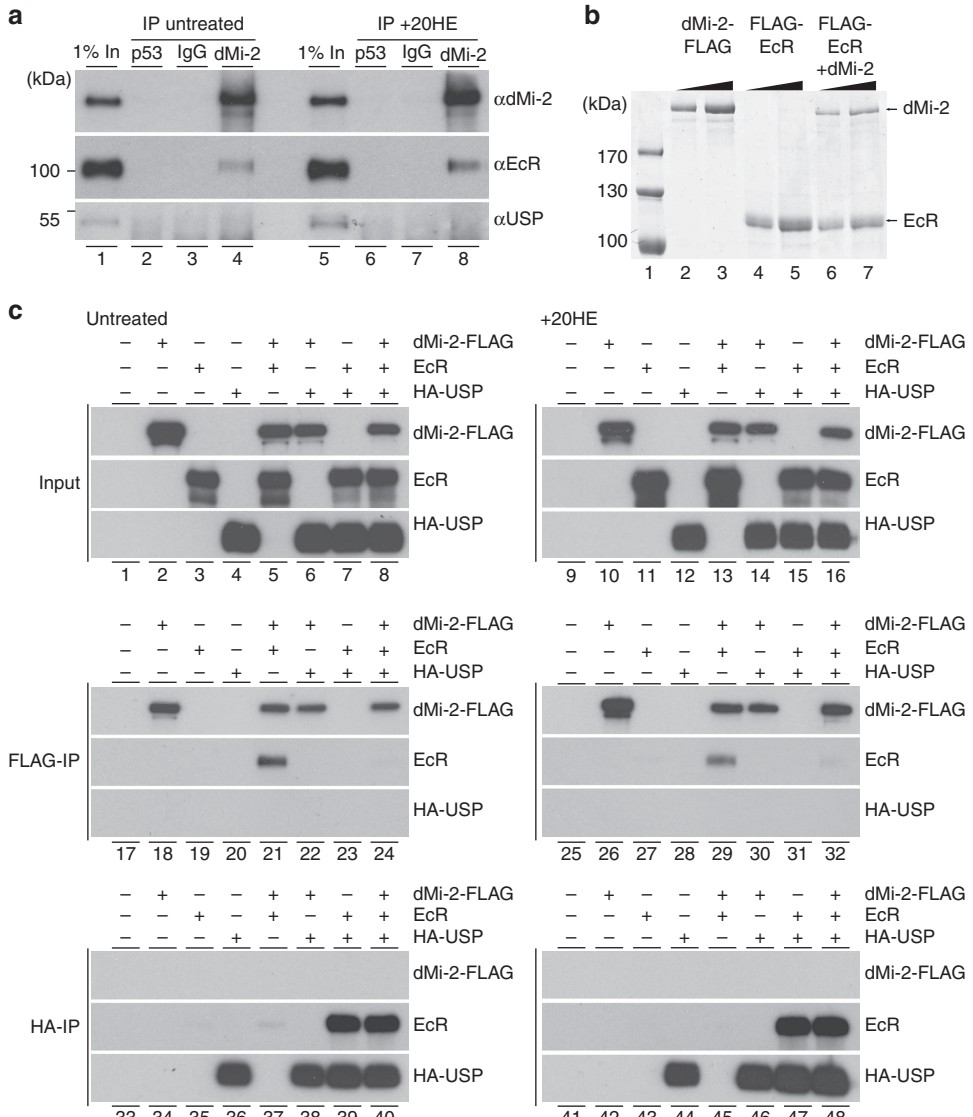

**Figure 4 | dMi-2 forms a complex with EcR and competes with USP for binding to EcR.** (**a**) dMi-2 and EcR interact. Nuclear extracts from untreated and ecdysone exposed ( + 20HE) S2 cells were immunoprecipitated with dMi-2 antibody, dp53 antibody or IgG as indicated on top (lanes 2–4 and 6–8). 1% input was loaded in lanes 1 and 5. Antibodies used for western blot analysis are indicated on the right, molecular masses on the left. (**b**) Sf9 cells were infected with recombinant baculoviruses directing the expression of dMi-2-FLAG, dMi-2 or EcR-FLAG as indicated on top. Extracts were immunoprecipitated with FLAG antibody and washed with high salt buffer. Immunoprecipitates were then analysed by SDS–PAGE and Coomassie staining. Lane 1: molecular weight marker. Lanes 2, 4 and 6: 500 ng protein, lanes 3, 5 and 7: 1 µg protein. (**c**) dMi-2 and USP bind EcR in a mutually exclusive manner. Sf9 cells that were either left untreated (left panels) or were exposed to ecdysone ( + 20HE, right panels) were infected with recombinant baculoviruses expressing dMi-2-FLAG, EcR or HA-USP as indicated on top. Extracts were analysed by western blot for expression of recombinant proteins (top panels). Extracts were immunoprecipitated with FLAG (FLAG IP, middle panels) or HA (HA IP, bottom panels) antibody and immunoprecipitates were analysed by western blot. Recombinant proteins detected by western blots are indicated on the right.

dMi-2 immunoprecipitate (middle left panel, lane 24). These results indicate that USP and dMi-2 compete for binding to EcR.

To determine the effect of ecdysone on these interactions, we repeated the interaction assays using infected Sf9 cells that were exposed to hormone during recombinant protein expression. This did not change the results (Fig. 4c, right panels). These findings confirm that EcR and USP heterodimerize in the absence of hormone[19] and correlate with our observation that dMi-2 and EcR coimmunoprecipitate from S2 extracts in a hormone-independent manner (Fig. 4a). To map which region of EcR binds to dMi-2, we generated baculoviruses expressing EcR mutants and tested their interaction with dMi-2 (Fig. 5a). This mapped the dMi-2 interaction domain to the LBD/AF2

domain of EcR that also serves as the binding interface for heterodimerization with USP[20].

Taken together, these results demonstrate that (i) dMi-2 binds EcR but not USP and (ii) dMi-2 and USP contact the same interaction domain on EcR. Simultaneous binding of dMi-2 and USP to EcR is possibly precluded by steric hindrance.

**The LBD/AF2 domain of EcR binds the ATPase domain of dMi-2.** We next mapped the EcR interaction domain on dMi-2 by glutathione S-transferase (GST) pulldown assay. We compared binding of *in vitro* translated EcR to GST-dMi-2 fusion proteins (Fig. 5b). Previous work has established that the N- and

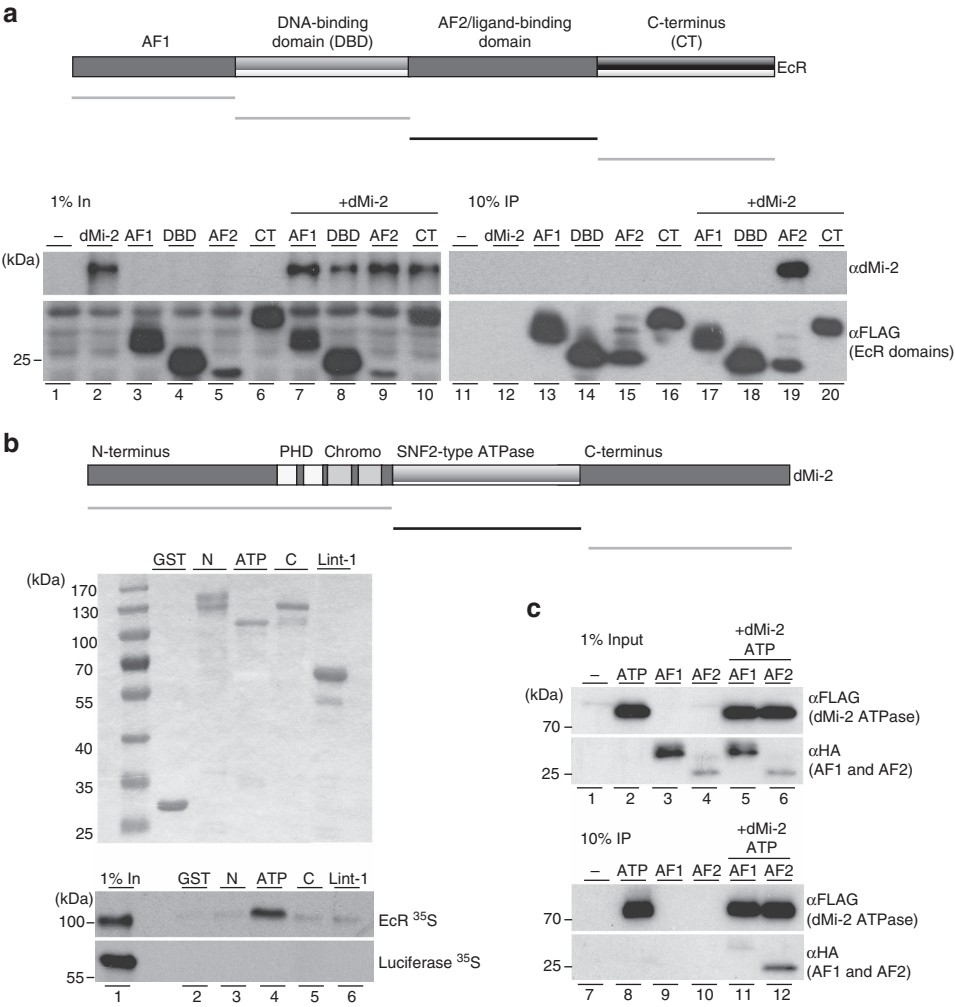

**Figure 5 | EcR LBD/AF2 domain and dMi-2 ATPase domain interact.** (**a**) Top: Schematic representation of EcR domain structure and deletion constructs used for interaction assay. Bottom: Sf9 cells were infected with baculoviruses expressing dMi-2 and FLAG-tagged EcR domains as indicated on top. Cell extracts were analysed by western blot for the expression of recombinant proteins (left panel, 1% input). Extracts were immunoprecipitated with FLAG antibody and immunoprecipitates were analysed by western blot (right panel, 10% IP). Antibodies used are indicated on the right, molecular masses are shown on the left. (**b**) Top: Schematic representation of dMi-2 domain structure and deletion constructs used for interaction assay. Middle panel: Coommassie-stained SDS–PAGE gel of GST constructs used. Bottom panel: Autoradiography of SDS–PAGE gel showing [35]S-EcR (top half) and [35]S-Luciferase (bottom half) bound to GST constructs. Lane 1: 1% input. Molecular masses are indicated on the left. (**c**) Sf9 cells were infected with baculoviruses expressing the FLAG-tagged ATPase domain of dMi-2, the HA-tagged AF1 domain of EcR or the HA-tagged LBD/AF2 domain of EcR as indicated on top. Cell extracts were analysed by western blot for expression of recombinant proteins (top panel, 1% input). Extracts were immunoprecipitated with FLAG antibody and immunoprecipitates were analysed by western blot (bottom panel, 10% IP). Antibodies used are indicated on the right, molecular masses are shown on the left.

C-terminal regions of CHD4 and dMi-2 serve as nucleosome or protein interaction surfaces while the ATPase domain is used for catalytic functions[15,21–26]. Surprisingly, N- and C-terminal regions showed only weak EcR-binding activity (bottom panel, lanes 3 and 5) that was comparable to background binding exhibited by an unrelated control protein (Lint-1 CT, lane 6). By contrast, EcR bound strongly to the ATPase domain (lane 4). None of the GST fusion proteins interacted with luciferase, further demonstrating that the observed interaction is specific. We then tested whether the isolated LBD/AF2 domain of EcR and the ATPase domain of dMi-2 are sufficient for interaction. We coexpressed FLAG-tagged dMi-2 ATPase domain and EcR domains AF1 and LBD/AF2 in Sf9 cells, immunoprecipitated dMi-2 and monitored interactions by western blot (Fig. 5c). Whereas binding of AF1 was barely detectable (lane 11), the LBD/AF2 efficiently interacted with the ATPase domain

(compare lane 6 (input) with lane 12 (IP)). This demonstrates that LBD/AF2 and the ATPase domain are sufficient to mediate a stable interaction between EcR and dMi-2.

**EcR increases dMi-2-mediated nucleosome remodelling *in vitro*.** It is surprising that EcR directly contacts the dMi-2 ATPase domain. We considered the possibility that EcR binding modulates dMi-2 remodelling activity. To test this hypothesis, we used the restriction enzyme accessibility assay (REA assay;[27,28]). Mononucleosomes were reconstituted on a 230 bp DNA fragment containing a so-called '601' nucleosome positioning sequence. In this nucleosome, a restriction enzyme cleavage site is protected from digestion. Remodelling or sliding of the nucleosome makes this site accessible resulting in DNA cleavage by a restriction endonuclease added to the reaction (Fig. 6a). dMi-2 increases

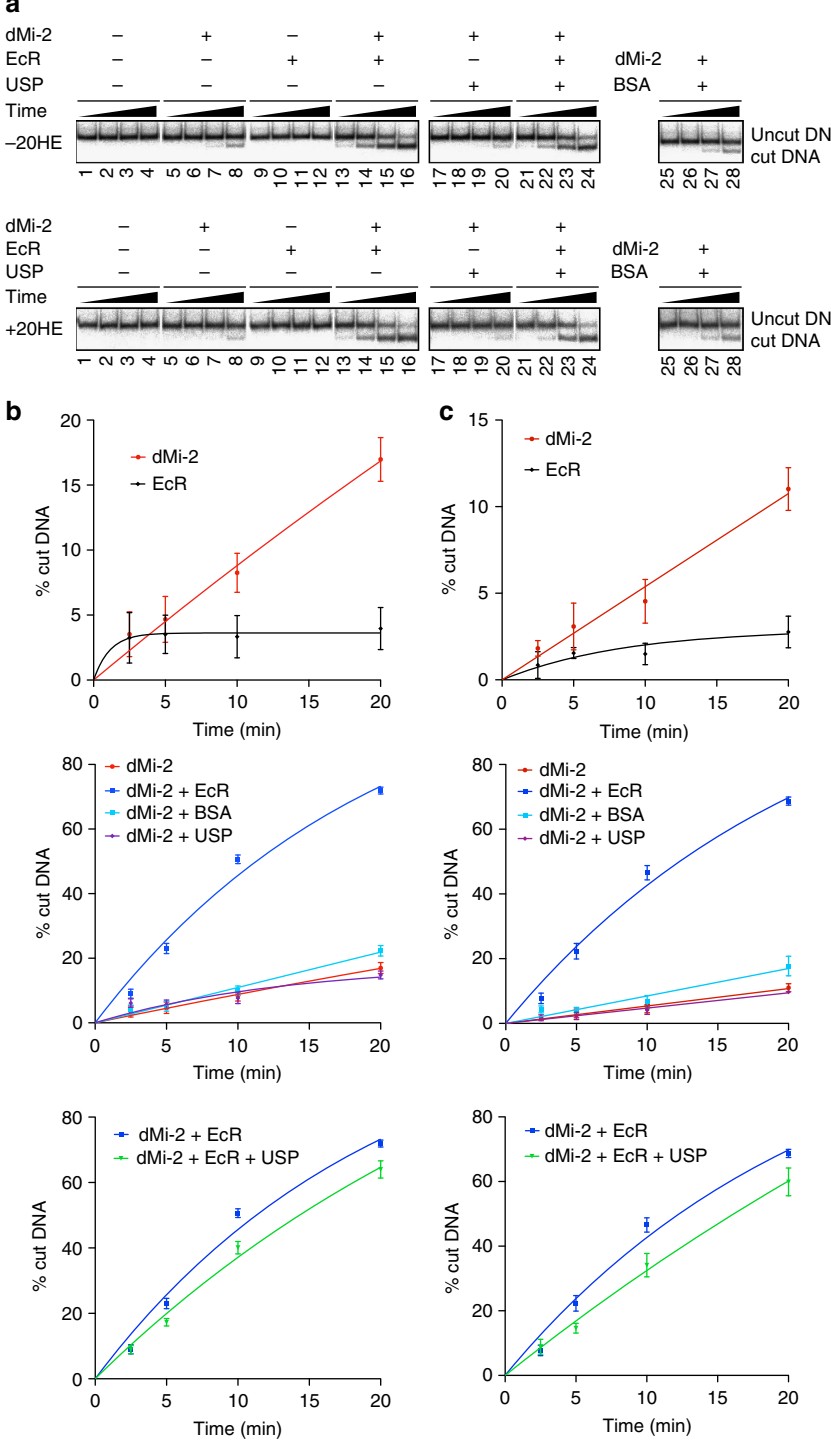

**Figure 6 | EcR increases dMi-2-mediated nucleosome remodelling *in vitro*.** (**a**) REA assays were carried out with a $^{32}$P-labelled mononucleosome substrate in the presence of recombinant dMi-2, EcR, USP and BSA as indicated on top. Reactions were stopped at four time points (2.5, 5, 10 and 20 min) and analysed by non-denaturing PAGE and autoradiography. The positions of the uncut DNA fragment and the cut DNA fragment (product of the remodelling reaction) are indicated on the right. The top panel shows REA assays carried out in the absence ( − 20HE), the bottom panel shows REA assays carried out in the presence of 1 μM ecdysone ( + 20HE). (**b,c**) REA assays were carried out in the presence of 110 nM dMi-2, 110 nM EcR, 445 nM USP and/or 745 nM BSA for 2.5, 5, 10 and 20 min as indicated. Panel (**b**) shows results obtained in the absence of ecdysone; panel (**c**) shows results obtained in the presence of 1 μM ecdysone. Bands containing cut and uncut DNA were quantified using the Science Lab Image Gauge (FUJIFILMS) software. The ratio of cut and total DNA (cut plus uncut) is plotted as '% cut DNA'. Curves were fitted using the GraphPad Prism software according to one-phase decay equation. Error bars represent s.e.m. and are derived from four (**b**) and three (**c**) independent experiments.

fragment cleavage indicating nucleosome remodelling (Fig. 6a, top panel: lanes 5–8; Fig. 6b, top panel). In contrast to dMi-2, EcR on its own did not remodel the nucleosome (Fig. 6a, top panel: lanes 9–12; Fig. 6b, top panel). However, addition of EcR to dMi-2 resulted in significant stimulation of remodelling (Fig. 6a, top panel: lanes 13–16; Fig. 6b, middle panel). By

contrast, the addition of USP or BSA did not increases the efficiency of dMi-2-mediated remodelling (Fig. 6a: lanes 17–20, lanes 25–28; Fig. 6b, middle panel) arguing that dMi-2 remodelling activity is specifically stimulated by EcR. Given that USP competes with dMi-2 for binding to EcR when all three proteins are coexpressed in Sf9 cells (Fig. 4c), we tested whether addition of USP diminishes the stimulation of dMi-2 remodelling activity by EcR (Fig. 6a: lanes 21–24; Fig. 6b, bottom panel). USP addition resulted in a modest but statistically significant reduction of dMi-2 activation by EcR. We performed the same series of remodelling experiments in the presence of 20HE (Fig. 6a, bottom panel, Fig. 6c). In agreement with our findings that dMi-2 binding to EcR and competition between dMi-2 and USP for binding to EcR is independent of 20HE (Fig. 4), we also did not find a significant effect of hormone in our *in vitro* nucleosome remodelling assays.

We conclude that EcR, but not USP, increases dMi-2-mediated nucleosome remodelling *in vitro*. Moreover, this effect is counteracted by USP, presumably due to competition between USP and dMi-2 for binding to EcR.

## Discussion

Our data suggest that EcR increases dMi-2 binding to ecdysone-induced genes. This is in agreement with previous work: Most EIMRs within *Br-C* and *vrille* overlap or are adjacent to experimentally determined EcR-binding regions[29,30]. Indeed, our bioinformatic analysis revealed that EcR–USP-binding motifs are greatly enriched within EIMRs. Most importantly, dMi-2 recruitment is abrogated by EcR depletion. Taken together with our finding that EcR physically binds dMi-2, our results strongly suggest that EcR recruits dMi-2 to chromatin through a direct interaction. Several of our results support the hypothesis that dMi-2 recruitment does not involve USP. First, we failed to detect USP in anti-dMi-2 immunoprecipitates that contain EcR. Second, overexpression of recombinant USP abrogates formation of an EcR-dMi-2 complex *in vitro*. Third, dMi-2 and USP bind to the same interaction domain on EcR. Fourth, depletion of EcR but not of USP reduces recruitment of dMi-2 to chromatin, suggesting that chromatin-bound EcR-dMi-2 complexes do not contain USP. EcR complexes that lack USP have previously been reported: EcR heterodimerizes with the orphan receptor Seven-Up[31]. However, it is unlikely that such a complex is involved in dMi-2 recruitment given that the LBD/AF2 domain is used as the EcR–Seven-Up heterodimerization interface and, therefore, would not be available for dMi-2 binding. In conclusion, our results suggest that dMi-2 regulates transcription as part of a non-canonical EcR–dMi-2 complex.

Ecdysone promotes formation of EcR–USP heterodimers and increases their nuclear localization and DNA binding[6]. Given that USP competes with dMi-2 for binding to EcR *in vitro*, it is surprising that dMi-2 binding to ecdysone-regulated genes increases in hormone-treated cells. We propose that ecdysone-regulated genes are not only occupied by EcR–USP heterodimers but, in addition, by EcR monomers and/or homodimers (that dimerize via their DNA binding domain) and that the number of chromatin-bound EcR monomers/homodimers increases upon hormone exposure. As these EcR molecules are fully capable of interacting with dMi-2, they have the potential to recruit dMi-2 to ecdysone-regulated genes. Analysis of mammalian nuclear receptor-mediated gene activation suggests that this is a dynamic process which entails continuous dissociation and re-association of transcription factor complexes[7,8,32,33]. It is conceivable that hormone-mediated EcR–USP binding to DNA followed by heterodimer dissociation will generate

chromatin-associated EcR monomers/homodimers that are available for dMi-2 binding and recruitment. Moreover, depending on the sequence of the binding site, EcR does not strictly require heterodimerization with USP for DNA binding but can function as a monomer/homodimer *in vitro* (Seibel, 1999 Diss. ETH No 13355). Interestingly, the human orphan receptor ROR gamma, which binds DNA as an obligatory monomer, interacts directly with the dMi-2 homologue CHD4 (ref. 34). This suggests that recruitment of the remodeller by nuclear hormone receptor monomers might be a more general principle. The notion that chromatin-bound EcR monomers/homodimers exist *in vivo* is supported by recent ChIP-seq studies, which have identified many EcR-associated genomic regions that are apparently devoid of USP[35]. Furthermore, USP-deficient animals respond to the mid-third instar ecdysone pulse with normal activation of *Br-C* transcription by EcR, suggesting that EcR can indeed regulate transcription outside of the canonical EcR–USP heterodimer[36].

Most known interactions of the EcR–USP heterodimer with coactivators or corepressors are governed by hormone binding. For example, ecdysone promotes formation of a complex containing EcR and the histone methyltransferase TRR and is required for NURF binding to EcR–USP[5,37]. Conversely, hormone abrogates the association of the corepressors SMRTR and Alien to the hormone receptor[4,38]. These results agree with the bimodal switch model which postulates that activation of hormone-dependent genes by nuclear hormone receptors is accompanied by an exchange of corepressors for coactivators. The complex formed by dMi-2 and EcR does not follow this general principle: Both endogenous and recombinant dMi-2 and EcR proteins interact irrespective of the presence of ecdysone, suggesting that their interaction is not regulated by hormone. Accordingly, dMi-2 is associated with ecdysone target genes and minimizes their basal transcription levels in the absence of hormone, when these genes are expected to be bound by unliganded EcR. The ability of dMi-2 to retain binding to EcR in the presence of hormone affords it with the potential to also modulate transcription during gene activation. Thus, the hormone-independent mode of the dMi-2-EcR interaction allows dMi-2 to constrain transcription in both scenarios. Our results show that dMi-2 acts as a corepressor of EcR but its modes of action are not adequately described by the bimodal switch model of corepressor and coactivator function.

Depletion of dMi-2 increases MNase accessibility at the *vrille* gene. Although we cannot exclude the involvement of non-histone factors, a plausible explanation for this effect is that dMi-2 positions nucleosomes over the *vrille* promoter to limit access of transcription factors and the transcription machinery to promoter DNA. This hypothesis is consistent with the ability of dMi-2 to remodel and reposition nucleosomes *in vitro*[39,40] and the propensity of dNuRD to increase histone density at its target sequences[41]. The increase in chromatin accessibility in dMi-2 depleted cells correlates with increased transcription arguing that dMi-2-mediated chromatin alterations help to limit the dynamic range of gene transcription. This is reminiscent of the role of CHD4 in the early mouse embryo where it limits the frequency of expression of lineage-specific genes[14]. Furthermore, dMi-2 extensively colocalizes with active RNA polymerase II on polytene chromosomes, suggesting that it constrains the transcription of many genes[40,42]. This might reflect a general property of this class of nucleosome remodellers. We note that the competitive and mutually exclusive binding of dMi-2 and USP to EcR could provide an additional repression mechanism: displacement of USP by dMi-2 has the potential to limit the number of EcR–USP heterodimers that can recruit coactivators.

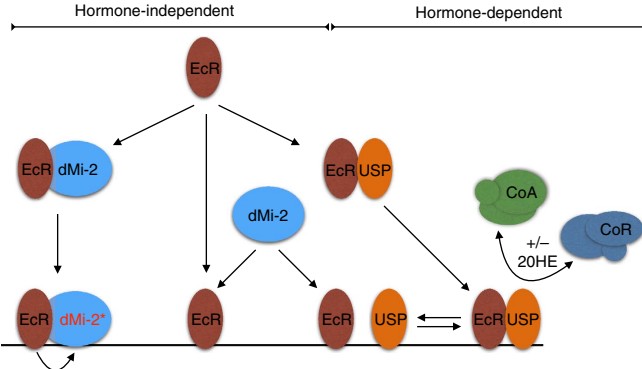

**Figure 7 | Model.** Model detailing hormone-dependent (bimodal switch model) and hormone-independent modes of gene regulation by EcR. EcR can heterodimerize with either dMi-2 or USP. EcR–USP heterodimers recruit corepressor (CoR) or coactivator (CoA) complexes in a hormone-dependent manner. dMi-2 recruitment is independent of hormone. dMi-2 binds to EcR monomers or homodimers in solution (left) or DNA-bound EcR monomers/homodimers that are generated by monomer/homodimer binding to DNA (middle) or by dissociation of DNA-bound EcR–USP heterodimer (right). EcR activates dMi-2 nucleosome remodelling activity.

The physical interaction between EcR and dMi-2 described in this study potentially goes beyond recruiting the remodeller to chromatin. It is possible that an EcR-bound nucleosome provides a better substrate for dMi-2-mediated remodelling than the nucleosome alone. However, we favour the hypothesis that the stimulation of dMi-2 nucleosome remodelling activity results from EcR contacting the ATPase domain of dMi-2. Previously, the activation domain of the GAL4-VP16 transcription factor has been demonstrated to redirect SWI/SNF complex-mediated nucleosome sliding towards nucleosome eviction *in vitro*[43]. However, it is not known whether this process involves a direct interaction between the transcription factor and the SWI2 ATPase domain. The mechanism of how EcR might stimulate dMi-2 remodelling is currently unclear and will require further investigation. Intramolecular inhibition of remodelling activity and relief from this inhibition by nucleosome binding has recently been identified as an important regulatory principle for nucleosome remodellers. The remodelling activity of Chd1 is inhibited by an intramolecular interaction between its chromodomains and its ATPase domain[44]. Chromodomain binding to nucleosomes disrupts this inhibitory interaction and stimulates ATPase and remodelling activity. ISWI remodelling activity is similarly repressed by an intramolecular interaction between the ISWI AutoN and ATPase domains[45]. In the latter case, inhibition is relieved when the enzyme binds histone H4 tails. These mechanisms ensure that Chd1 and ISWI acquire maximum activity when they encounter their cognate nucleosome substrates. It is conceivable that dMi-2 remodelling activity is similarly curtailed by an inhibitory intramolecular interaction involving its ATPase domain. Indeed, contacts between chromodomains and ATPase domain similar to those identified in Chd1 have been demonstrated for human Mi-2/CHD4 (ref. 26). Similar to Chd1 and ISWI, such an inhibitory intramolecular interaction in dMi-2 might also be disrupted by binding to a nucleosome substrate and, in addition, by interacting with the EcR.

Our results extend the bimodal switch model for EcR function (Fig. 7). The EcR–USP heterodimer provides hormone-dependent regulation of transcription as postulated by the bimodal switch model. In addition, the alternative EcR-dMi-2 complex constrains transcription in a hormone-independent manner. We propose

that EcR-dMi-2 complexes form in the nucleoplasm and then bind to DNA or that EcR binds to DNA as a monomer/dimer followed by dMi-2 recruitment. Also, dissociation of EcR–USP heterodimers on chromatin provides additional EcR monomers capable of recruiting Mi-2. In the presence of hormone, more EcR and USP enter the nucleus and the amount of DNA-bound EcR–USP increases[6]. This in turn would be expected to increase the number of DNA-bound EcR monomers resulting in increased dMi-2 chromatin association and thus would prevent an excessive transcriptional response.

The results presented in our study reveal an unanticipated dynamic interplay between EcR and the nucleosome remodeller dMi-2 that involves formation of a non-canonical EcR-corepressor complex, recruitment of dMi-2 to chromatin and direct activation of its remodelling activity. The finding that the monomeric nuclear hormone receptor ROR gamma, which plays important roles in regulating mammalian development, also directly binds to CHD4 (ref. 34) suggests that the molecular mechanisms revealed in this study may have a broader significance.

## Methods

**Cell and baculovirus culture.** S2 and Sf9 cell lines (kind gift from Peter Becker, Munich) were maintained at 26 °C in Schneider medium (Gibco) and Sf-900 medium (Gibco), respectively, supplemented with 10% fetal calf serum. RNA interference, baculovirus generation and infection are described in ref. 40. Briefly, double-stranded RNA was generated by T7 Polymerase *in vitro* transcription from PCR ampliers generated with T7 promotor-containing primers (Supplementary Table 1). Double-stranded RNAs were transfected into S2 cells using Effectene (Qiagen). Baculoviruses were generated using the Bac-to-bac system (Invitrogen). Baculoviruses were amplified twice and then used to infect Sf9 cells for protein production. Cells were then harvested 48–72 h after infection. EcR (ER33854) and USP (LD09973) cDNAs were obtained from BDGP. Vectors for generation of baculoviruses expressing untagged EcR, N-terminally FLAG-tagged EcR and N-terminally HA-tagged USP were generated by PCR-cloning of the respective open-reading frames into pFastBac or pVL1392 using appropriate sets of primers. Baculoviruses and expression vectors for dLint-1 and dMi-2 were constructed in the same manner[46,47].

**Western blot.** Western blots were carried out by speparating proteins with SDS–polyacrylamide gel electrophoresis (SDS–PAGE) and electroblotting onto activated polyvinylidene difluoride membranes in Blotting Buffer (20 mM Tris, 192 mM glycin, 20% methanol, 0.02% SDS). Membranes were then incubated in Blocking Buffer (PBS, 0.1% Tween 20, 5% non-fat dry milk) for 1 h at room temperature followed by an overnight incubation in blocking buffer with appropriate primary antibody (see below) at 4 °C. Membranes were washed three times for 5 min in Washing Buffer (PBS, 0.1% Tween 20) and then incubated in blocking buffer containing the appropriate secondary antibody Anti-Mouse IgG (Horseradish Peroxidase-Linked Species-Specific Whole Antibody from sheep; Amersham, NA931; 1:20.000) or Anti-rabbit IgG (Horseradish Peroxidase-Linked Species-Specific Whole Antibody, from donkey; Amersham, NA934; 1:20.000) for 2 h at room temperature before a final wash cycle (3 times, 5 min). Western blot signals were visualized by chemiluminescence using the Immobilon Western Chemiluminescence HRP substrate (Millipore, WBKLS0500). The following primary antibodies were used: Rabbit polyclonal anti-dMi-2 antibody (custom made, 10 mg ml$^{-1}$; 1:10,000), beta-Tubulin antibody (KMX-1; Millipore, MAB3408, 0.5 mg ml$^{-1}$; 1:15,000), FLAG (Sigma, F7425, 8 mg ml$^{-1}$; 1:5,000), USP antibody (Abcam, ab106341, 0.2 mg ml$^{-1}$; 1:2,000) and EcR antibody (DHSB, DDA2.7, 0.014 mg ml$^{-1}$; 1:1,000). Uncropped versions of western blots are shown in Supplementary Fig. 4.

**Preparation of recombinant proteins and interaction assays.** GST fusion proteins were expressed from pGEX4T1 expression vectors in frame with a N-terminal GST-tag. Vectors were transformed into *Escherichia coli* BL21. In all, 500 ml cultures in liquid medium were incubated at 37 °C to an OD$_{600}$ of 0.6–0.7. Temperature was reduced to 18 °C prior to induction with 0.1 mM IPTG. Expression was continued overnight. Cells were harvested by centrifugation (1,000 g, Heraeus Cryofuge 5,000) and resuspended in 30 ml PBS containing 1% (v/v) Triton X-100. After sonication (10× 12 s, 25% output), cell debris was pelleted by centrifugation (30 min at 26,800g at 4 °C Sorvall RC-5B, SS34 rotor). Clear supernatant was bound to 500 μl of prewashed Glutathione Sepharose 4 Fast Flow (GE Healthcare) for 2 h on a rotating wheel at 4 °C. Beads were washed five times with 10 ml PBS containing 1% (v/v) Triton X-100, resuspended in PBS containing 40% (v/v) Glycerol and stored at − 20 °C. $^{35}$S-labelled EcR was synthesized using the TNT Quick Coupled Reticulocyte Transcription/Translation

System (Promega). Full-length EcR and EcR fragments were cloned into the pING14A vector under control of a T7 RNA polymerase promoter. The IVT reaction (12.5 µl rabbit reticulocyte lysate, 1 µl reaction buffer, 0.5 µl T7 RNA polymerase, 0.5 µl amino acid mixture without methionine, 1 µl $^{35}$S-methionine ($>$1,000 Ci per mmol at 10 mCi per ml, Hartmann Analytic), 0.5 µl RiboLock RNase Inhibitor (Thermo Scientific), 2 µl DNA template (0.5 µg per µl) and 7 µl nuclease-free double distilled water (Ambion)) was incubated at 30 °C for 90 min.

The reaction was then diluted in GST pulldown buffer (25 mM Hepes, pH 7.6, 150 mM NaCl, 12.5 mM MgCl₂ 0.1% (v/v) NP-40, 0.1 mM DTT) and incubated with 2 µg GST-fusion protein bound to glutathione beads for 2 h at 4 °C on a rotating wheel. Beads were collected by centrifugation and washed five times with 1 ml of GST pulldown buffer for 5 min at 4 °C. Proteins were eluted by boiling in SDS–PAGE loading buffer and subjected to SDS–PAGE. Gels were fixed with fixing solution (25% (v/v) isopropanol, 10% (v/v) acetic acid), treated with Amplify (GE Healthcare) for 30 min at room temperature and dried. Dried gels were exposed to a SuperRX Fuji Medical X-ray film.

Whole-cell lysates of infected Sf9 were prepared by three freeze/thaw cycles in Lysis Buffer (20 mM Hepes, pH 7.6, 200 mM KCl, 0.1% NP40, 10% glycerol). Lysates were cleared by centrifugation. Immunoprecipitation of FLAG- and HA-tagged proteins from these lysates were carried out using immobilized antibodies (100 µl of 8 mg per ml FLAG agarose (Sigma, A2220), 100 µl of 2.1 mg per ml HA agarose (Sigma, A2095))[16,47].

Antibody beads were incubated with Sf9 whole-cell extracts for 4 h at 4 °C. Beads were then washed five times with Lysis Buffer. For copurification of EcR-dMi-2 complexes, FLAG agarose beads were washed with high salt buffer instead (20 mM Hepes, pH 7.6, 1000 mM KCl, 0.1% NP40, 10% glycerol).

**Nuclear extract preparation and immunoprecipitations.** S2 cells were harvested, washed twice in PBS and resuspended in an appropriate volume (1 ml per 75 cm² flask) of low salt buffer (10 mM Hepes KOH, pH 7.6, 1.5 mM MgCl₂, 10 mM KCl, 0.1 mM DTT). After incubation on ice for 10 min, cells were collected by centrifugation at 14,800g for 1 min at 4 °C. The supernatant was removed and the remaining nuclear fraction was resuspended in an appropriate volume (200 µl per 75 cm² flask) of high salt buffer (20 mM Hepes KOH, pH 7.6, 1.5 mM MgCl₂ 420 mM NaCl, 0.2 mM EDTA, 20% (v/v) Glycerol, 0.1 mM DTT). The suspension was incubated for 20 min on ice and subsequently centrifuged at 14,800g for 30 min at 4 °C. The supernatant (nuclear extract) was aliquotted, frozen in liquid nitrogen and stored at −80 °C (refs 16,47). For immunoprecipitation of endogenous Mi-2, Protein G Sepharose was preincubated for 1 h with 1% fish skin gelatin and buffer D (0.2 mg per ml BSA in 20 mM Hepes KOH, pH 7.6 100 mM KCl 1.5 mM MgCl₂, 0.2 mM EDTA, 20% (v/v) glycerol, 0.1 mM DTT, protease inhibitors) at room temperature. In all, 2 µg of appropriate antibody (rat monoclonal anti-dMi-2 antibody (4D8; custom made, 0.5 mg per ml) or rat monoclonal dp53 antibody (7A11; custom made, 0.5 mg per ml) incubated with preblocked Protein G sepharose for 1 h at 4 °C. Antibody-loaded beads were washed twice with buffer D containing 0.02% NP40 and transferred to low-binding tubes. A total of 30 µl of antibody-loaded beads were then incubated with 500 µg nuclear extract for 3 h at 4 °C. The beads were washed five times with buffer D containing 0.02% NP40, eluted with 40 µl 2 × SDS loading buffer and subjected to SDS–PAGE and western blot.

**qRT-PCR.** Total RNA from S2 cells was isolated using the peqGOLD total RNA kit (Peqlab). In all, 1.5 mg of RNA was reverse transcribed with 0.5 mg Oligo(T)17 primer and 100 U M-MLV reverse transcriptase (Invitrogen). cDNA was analysed by qPCR using gene-specific primers (Supplementary Table 2). All amplifications were performed in triplicates. Triplicate mean values were calculated according to the DDCT quantification method using Rp49 transcription as normalization reference. S.d. were calculated from triplicates, error bars are indicated accordingly. Relative mRNA levels in GFP RNAi-treated S2 cells were set to 1 and other values were expressed relative to this.

**ChIP-qPCR and ChIP-Seq.** ChIP-qPCRs were performed by crosslinking of cells with formaldehyde (1% v/v), lysis, sonication, reversal of crosslinks, DNA purification and quantitative PCR[47]. A total of $10^8$ S2 cells in culture medium were fixed with 1% (v/v) formaldehyde (10% methanol free stock, Polysciences, 04018-1) for 10 min at room temperature (RT). Fixation was quenched with a final concentration of 240 mM glycine. Cells were harvested, washed twice in ice cold PBS, resuspended in 1 ml ChIP lysis buffer (50 mM Tris, pH 8.0, 10 mM EDTA, 1% (w/v) SDS) and incubated for 10 min on ice. Samples were sonicated with a Bioruptor (Diagenode) twice for 10 min with 30 s on–off cycles at high power. Samples were centrifuged at 14,800g for 15 min at 4 °C. The supernatant (chromatin) was diluted 10-fold in ChIP IP buffer (16.7 mM Tris, pH 8.0, 16.7 mM NaCl, 1.2 mM EDTA, 0.01% (w/v) SDS, 1.1% (w/v) Triton X-100). In all, 130 µl of chromatin were diluted 1:10 in ChIP IP buffer and precleared by the addition of 80 µl Protein A beads (GE Healthcare) for 30 min at 4 °C on a rotating wheel. The supernatant was collected, 13 µl were removed and stored at 4 °C (input control). A total of 2 µl rabbit polyclonal anti-dMi-2 antibody (custom made, 10 mg per ml) was added and the sample was incubated overnight at 4 °C on a rotating wheel. In all, 35 µl of 1:1 slurry of Protein A beads was added and incubation was continued for 2 h at 4 °C on a rotating wheel. The sample were repeatedly washed (three times with low salt buffer (20 mM Tris, pH 8.0, 2 mM EDTA, 0.1% (w/v) SDS, 1% (v/v) Triton X-100), three times with high salt buffer (same as low salt buffer plus 500 mM NaCl), once with LiCl buffer (10 mM Tris, pH 8.0, 250 mM LiCl, 1 mM EDTA, 1% (w/v) SDS, 1% (v/v) NP-40 and twice with TE buffer (10 mM Tris, pH 8.0, 0.1 mM EDTA)) for 10 min on a rotating wheel and the beads were collected by centrifugation at 1,200g for 4 min. With the last TE buffer wash, beads were transferred into fresh reaction tubes for elution. Elution was performed twice with 250 µl ChIP elution buffer (1% (w/v) SDS, 0.1 M NaHCO₃) for 20 min at RT. Crosslinks were reversed the addition of 20 µl of 5 M NaCl and incubation at 65 °C overnight. Proteins were digested with 2 µl Proteinase K (Roth, 7528.1; 10 mg per ml) in 20 µl 1 M Tris, pH 6.5 for 1 h a 45 °C. Precipitated DNA was purified using the QIAquick PCR Purification Kit (Qiagen). See Supplementary Table 3 for ChIP-qPCR primers. ChIP–Seq was carried out on an Illumina Genome Analyzer IIx according to the manufacturer's instructions. Raw Illumina sequence reads were counted using a bloom filter and aligned to the D. melanogaster genome (Ensembl 75) with Bowtie 2 version 2.0.0-beta7 (ref. 48) using default options, yielding 6,732,092 and 9,422,152 usable reads for two ' + Ecdysone' replicates and 14,989,665 and 14,992,427 usable reads for two '− Ecdysone' replicates. Peak calling was performed with MACS[49] (1.4.0rc2 20110214 (Valentine)) using the settings: non-default mfold = 8.30 and off-auto = True. Gene annotation was obtained from Ensembl revision 75. Transcription start sites were extracted from Ensembl transcript annotations to include internal transcription start sites. For normalization of lanes, read counts were normalized to 1 million uniquely mapping reads and peaks were classified as different when they had a ' + Ecdysone'/'− Ecdysone' tag count ratio of at least 2.3. Furthermore, peaks were considered to overlap when they shared at least 1 bp.

**MNase analysis.** The MNase protection assay was performed as described in ref. 50. Briefly, cells were crosslinked by the addition of 10 × MNase cross-linking buffer (50 mM Tris pH 8.0, 100 mM NaCl, 1 mM EDTA, 0.5 mM EGTA, 3.3% (v/v) methanol-free formaldehyde) to a final concentration of 1 × for 1 min at room temperature. The reaction was quenched by the addition of glycine (final concentration 125 mM) and chromatin was prepared. Chromatin was digested with MNase. The amount of MNase necessary to digest most of the chromatin to mononucleosomes was determined empirically for each new batch of MNase by titration. See Supplementary Table 4 for qPCR primers used. All qPCR values were normalized to values obtained with a corresponding undigested sample.

**Restriction enzyme accessibility assay.** REA assay was carried out as described[28]. Briefly, remodelling reactions were carried out on a $^{32}$P-labelled mononucleosome. In this mononucleosome, the histone octamer occupies a 601-positioning sequence containing a MfeI restriction site. The nucleosome protects this site from digestion by MfeI. REA reactions in the presence of MfeI were initiated by the addition of $^{32}$P-labelled nucleosomes. Aliquots were removed at various times and quenched in 1.5 volumes of 10% glycerol, 70 mM EDTA, 20 mM Tris (pH 7.7), 2% SDS, 0.2 mg per ml xylene cyanole and bromophenol blue. The samples were deproteinized by proteinase K digestion and DNA fragments were separated on native polyacrylamide gels.

**Immunofluorescence.** Drosophila larvae were cultured at 26 °C. Third instar larvae were washed and salivary glands were dissected in PBS. Dissected glands were fixed for 5 min at room temperature in Fixing Solution (45% acetic acid, 1% formaldehyde) on a siliconized cover slip. Each glass slide was immediately frozen in liquid nitrogen, the coverslip was removed with a scalpel and the glass slide was collected into a Coplin jar prefilled with PBS. Collected glass slides were washed with PBS for 10 min while rotating. PBS was replaced by blocking solution (5% non-fat dry milk in PBS) and gentle rotation was continued for 30 min. Slides were rinsed in PBS, placed in a humid chamber and squashed polytene chromosomes were covered with 40 µl of primary antibody (rat monoclonal anti-dMi-2 (4D8; custom made, 0.5 mg ml$^{-1}$; 1:200) or rabbit anti-ISWI antibody (kind gift from Carl Wu, 10 mg ml$^{-1}$; 1:200)[18,46] and a fresh cover slip. All antibodies were diluted in 5% milk in PBS and 2% normal goat serum to reduce unspecific binding of antibodies. Primary antibodies were incubated overnight at 40 °C. Glass slides were rinsed in PBS and washed three times in 5% milk in PBS for 5 min. Polytene chromosomes were incubated with the appropriate secondary antibodies (anti-rabbit Alexa488; 1:200 (Invitrogen) or anti-rat Alexa546; 1:200 (Invitrogen)) diluted in 5% milk/PBS/2% NGS for 1 h at RT in the dark. Slides were washed twice for 10 min with Buffer A (PBS plus 300 mM NaCl, 0.2% (v/v) NP-40, 0.2% (v/v) Tween-20) and B (PBS plus 400 mM NaCl, 0.2% (v/v) NP-40, 0.2% (v/v) Tween-20), rinsed in PBS and DNA was stained with DAPI (0.2 µg ml$^{-1}$ in PBS) for 4–5 min. Slides were washed once for 10 min in PBS, mounted with Fluoromount, sailed with nail polish and stored at 4 °C in the dark.

**Data availability.** ChIP-seq data are available at www.ebi.ac.uk/arrayexpress/experiments/E-MTAB-4577, under the accession code E-MTAB-4577. All other data are available from the authors upon reasonable request.

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

## Acknowledgements

This work was supported by funding from the Deutsche Forschungsgemeinschaft (IRTG 1384 and TRR81). We thank Guntram Suske for critical reading of the manuscript and Carl Wu for the gift of ISWI antibody.

## Author contributions

J.K., K.B., K.K., I.M., A.L.E., E.E., M.M. and I.U. performed experiments; F.F. and R.P. analysed ChIP-seq data; K.B. and A.B. designed experiments and wrote the manuscript.

## Additional information

**Competing interests:** The authors declare no competing financial interests.

