## [Peer Review File · Nature Communications]

Reviewers' comments:

Reviewer #1 (Remarks to the Author):

The manuscript by Kreher et al. analyzed the interaction of the ecdysone receptor (EcR), a member of the nuclear hormone receptor family, with the chromatin remodelling factor dMi-2. Authors identified the co-recruitment of dMi-2 to chromatin binding sites of EcR, an enhanced recruitment of dMi-2 by ecdysone, a direct interaction of the EcR with dMi-2 and suggest a competition between dMi-2 with the heterodimer partner of the EcR ultraspiracle, USP, for binding to the EcR. Interestingly, authors show an enhanced remodelling activity of dMi-2 in the presence of EcR. Knockdown of dMi-2 enhances ecdysone-mediated gene induction, suggesting that Mi-2 is mediating gene repression in the absence and presence of the cognate ligand. A further novelty is provided by the knockdown data that suggest that on those genes analyzed USP seems to be dispensable for EcR-mediated gene regulation.

In general the manuscript is very interesting, very original and has important novelties. The quality of presentation of the data is very high and the manuscript clearly written. However some conclusions require further supportive evidence, statistics and some additional controls. Chromatin remodelling factors have been suggested earlier to be important for both gene activation and repression of nuclear hormone receptors in the absence or presence of ligand (reviewed by Collingwood et al., 1999). Also ligand-dependent HDAC and corepressor recruitment to nuclear hormone receptors has been proposed. However, the impact of nuclear receptors on chromatin remodelling activity has not yet been shown. Also the role of USP in chromatin remodelling by EcR is very novel.

Some points will improve the quality of the manuscript:

Major points:

1. Authors suggest that USP is hindering dMi-2 to bind to EcR. Thus either the depletion of USP might enhance or the overexpression might inhibit dMi-2 mediated remodelling activity at EcR sites. Authors should verify their conclusions functionally by one of these points in S2-cells.

2. The inhibitory effect of USP on dMi-2 activity is not convincing. Essential for any conclusion of the data in Fig. 6 is the statistics. Therefore Fig. 6D requires statistical analysis of significance.

3. The data of Fig. 3C further suggest that USP is dispensable for 20HE induced gene expression at a specific time point. It is however possible that a highly timely dynamic recruitment of factors occur at the promoters. To confirm that USP is dispensable, RNAi and time-dependent experiments should be performed similar to that of Fig. 2B and 2C.

- Please indicate in the figure legend the treatment time and concentration of 20HE.

- Furthermore, the derepression experiments shown in Fig. 2D and 2E of Br-Rb and vril1 as well as that of the two lncRNAs should include the RNAi experiments for USP.

- Authors should knockdown USP and analyze the expression of various other EcR responsive genes in the absence and presence of 20HE to generalise the conclusions.

4. Analyzing the data points of Fig. 6C and 6D there are rather sigmoidal and not a linearity of the data obtained with incubation of Mi-2 and EcR mediated remodelling activity as shown in Fig. 6C and Fig. 6D. What is the evidence to exclude a sigmoid interdependency? Authors should discuss the possibility of a sigmoid curve / interdependency.

5. Please indicate the role of 20HE in Fig. 6 and show results in the presence of 20HE.

6. The data suggest that EcR binds to the ATP binding site of dMi-2 and activates its remodelling activity. Please show whether this depends on ATP concentrations using the same experimental setup shown in Fig. 6D. Were saturating ATP concentrations used?

7. Authors suggest that an EcR monomer is binding to dMi-2 on DNA. The affinity of a monomer of EcR might be too weak in vivo at chromatinized sites. Nuclear receptors are able to also dimerize with their DNA binding domain. What is the evidence that authors exclude a homodimer of EcR? If no clear proof of a monomeric EcR is provided authors should change their statements and the Fig. 7.

Minor points:

8. Depict in Fig. 5A and 5B the linearity of both factors since the migration of the bands does not reflect the protein regions indicated in the schemes.

9. Indicate in Fig. 1B legend the treatment time of 20HE.

Reviewer #2 (Remarks to the Author):

Ecdysone receptor (EcR) and Ultraspiracle (USP) regulate key developmental gene expression by interactions with corepressors or coactivators. This study shows that the nucleosome remodeler dMi-2 is recruited to ecdysone-regulated genes to limit basal transcription. Interestingly, recruitment of dMi-2 corepressor is increased upon hormone addition to constrain gene activation through remodeling of chromatin, apparently through stimulation of dMi-2 ATPase and remodeling activities. This study reveals a non-canonical EcR-dMi2 complex and exposes a direct regulation of ATP-dependent remodeling activity by a nuclear hormone receptor. These findings add much details to the current bimodal switch model of gene regulation by nuclear hormone receptors and the exchange of corepressors and coactivators upon hormone stimulation.

Immunofluorescence study presented in this study demonstrated clear accumulation of dMi-2 at ecdysone-activated polytene chromosome puffs. ChIPseq identified classical ecdysone target genes as targets of dMi-2 binding in response to ecdysone treatment. RNA interference and RT-qPCR analyses show apparent constrain of transcription upon dMi-2 recruitment, and micrococcal nuclease digestion clearly show that dMi-2 is required for maintaining a closed chromatin conformation. Knockdown and ChIP experiments revealed that dMi-2 recruitment depends only on the EcR subunit. Furthermore biochemical analysis clearly demonstrated the formation of a EcR-dMi-2 complex that is devoid of USP, and USP completes with dMi-2 binding by interacting with the same region on EcR. Most interesting and importantly, EcR was shown to contact the ATPase domain of dMi-2 and can stimulate dMi-2 mediated nucleosome remodeling.

This study has presented convincing evidence that support important findings and advance in our understanding of transcriptional regulation by nuclear hormone receptors.

Reviewer #3 (Remarks to the Author):

This manuscript presents data indicating that dMi-2 is recruited to ecdysone regulated genes via interactions with EcR but not to USP. The manuscript also shows that that dMi-2 recruitment increases upon hormone stimulation and acts to constrain gene activation. This represents an important new insight in comparison to previous studies suggesting that the role of dMi-2 was to maintain the inactive state.

One of the most interesting aspects of the manuscripts is that it identifies contacts between EcR and the ATPase domain of dMi-2. Effects on the activity of dMi-2 are shown using a restriction enzyme accessibility assay using a 230 bp DNA fragment. It's possible that EcR could bind this DNA non-specifically and affect the outcome of ATP-dependent reactions. As a result I don't think

it is safe to conclude that EcR stimulates activity from this assay. It would be compelling if the authors could show an effect on ATPase activity as this would be independent of any effect EcR has on the template.

An alternative explanation is that the interaction between EcR and the ATPase domain acts to recruit dMi-2 rather than increase its turnover. This would be a more conventional mode of action and in this case experiments could be performed using templates with and without EcR binding sites.

Changes to chromatin are assessed at dMi-2 dependent genes by MNase digestion and PCR. It would be much more powerful to use MNase-seq as this would provide information about changes to nucleosome positioning at all EcR loci, rather than the evidence for generic changes to accessibility provided by the current data.

Reply to reviewers

We thank all three reviewers for their insightful and positive comments. Addressing the reviewer's concerns was delayed by our lab moving to a new campus and the associated problems with getting things up and running again. Nevertheless, we now have been able to conduct additional experiments and to incorporate most of the reviewer's suggestions. These include a more detailed study of the role played by USP in the regulation of ecdysone-dependent genes (new Figure 2), the inclusion of several additional genes in the analysis (new Figure 2), new remodeling assays aimed to better define the extent of EcR-mediated stimulation of dMi-2 and the effect of USP inhibition (new Figure 6), remodeling assays carried out in the presence of ecdysone (new Figure 6), a demonstration that ATP concentrations are saturating in the remodeling assays (Figure provided in this document) and an analysis of the effect of EcR on dMi-2 ATPase activity (Figure provided in this document). As a result, we feel that the manuscript has greatly improved.

Reviewer #1

(reviewer comments in *italics*)

We thank the reviewer for pointing out that our manuscript is "very original and has important novelties" and for praising the very high quality of the data.

Major points:

1. Authors suggest that USP is hindering dMi-2 to bind to EcR. Thus either the depletion of USP might enhance or the overexpression might inhibit dMi-2 mediated remodelling activity at EcR sites. Authors should verify their conclusions functionally by one of these points in S2-cells.

We appreciate the reviewer's suggestion to modulate USP levels in order to test effects on dMi-2 mediated remodeling activity at EcR sites. However, the inherent difficulty in interpreting any change in chromatin structure following EcR or USP knockdown (e.g. as measured by MNase accessibility) is that there is no way of telling whether this is a result of "dMi-2 mediated remodeling" or brought about by any other of the many chromatin modifying complexes that have been shown to interact with EcR/USP.

Modulating USP levels by overexpression/depletion in S2 cells would in principle be expected to cause changes in the dMi-2/EcR association. However, while dMi-2 can be efficiently displaced from EcR in vitro by recombinant USP expressed to very high levels using a viral expression system (Fig. 4C), we consider it unlikely that plasmid-driven overexpression of USP in S2 cells will be able to produce a similar black-and-white effect. Indeed, simply overexpressing USP or EcR in S2 cells is insufficient to effect changes in *Br-C* and *vriIIe* transcription (*data not shown*).

We found the reviewer's other suggestions to test the effects of USP depletion on gene transcription much better suited to illuminate the role of USP (see point 3 below).

2. The inhibitory effect of USP on dMi-2 activity is not convincing. Essential for any conclusion of the data in Fig. 6 is the statistics. Therefore Fig. 6D requires statistical analysis of significance.

We understand the concern of the reviewer given that the inhibition of EcR-stimulated dMi-2 remodeling by addition of USP is of a modest nature and the standard errors of the mean (SEM) reported do suggest significant experimental variation.

We now have repeated the remodeling assay many times and the results are robust over a range of different protein and nucleosome preparations. We include new data that combine the results from 3 replicates (new Figure 6; note that this figure now also includes new panels showing

remodeling reactions in the presence of ecdysone). Error bars denote SEM and demonstrate a reproducible and statistically significant reduction of remodeling activity.

The limited repression of EcR-stimulated dMi-2 remodeling by USP is in contrast to the efficient competition of the EcR/dMi-2 interaction by USP in the in vitro binding assay (Fig. 4C). However, the binding assay is carried out under stringent washing conditions where weakly/transiently bound molecules are irreversibly removed during washing steps whereas in the remodeling assay all three proteins remain present throughout the reaction, allowing them to bind, dissociate and rebind. Moreover, in the in vitro binding assay we can work with an excess of competing USP, the limitations of the in vitro nucleosome remodeling assay do not allow us to use similarly large quantities of USP. As a result, inhibition of EcR-stimulated dMi-2 remodeling by USP is less pronounced than inhibition of dMi-2 binding to EcR.

To better reflect the modest nature of inhibition of dMi-2/EcR remodeling by USP we have exchanged the statement:

“USP did not abolish stimulation completely but did decrease dMi-2 activation by EcR.”

by

“USP addition resulted in a modest but statistically significant reduction of dMi-2 activation by EcR.”

3. The data of Fig. 3C further suggest that USP is dispensable for 20HE induced gene expression at a specific time point. It is however possible that a highly timely dynamic recruitment of factors occur at the promoters. To confirm that USP is dispensable, RNAi and time-dependent experiments should be performed similar to that of Fig. 2B and 2C.

We agree with the reviewer that this is a possibility to consider. Figs 2B and 2C of the original manuscript show a 6 hour time course of ecdysone-activation of *Br-C* (Fig. 2B) and *vriIIe* (Fig. 2C) mRNA expression after knockdown of dMi-2, EcR and GFP (control) as measured by RT-qPCR. To determine the effects of USP knockdown during this time course we repeated these experiments this time including an USP knockdown. We have included the new data as new Figures 2A (showing knockdown efficiency as determined by Western blot), 2B (showing knockdown efficiency as determined by RT-qPCR), 2C (expression data for *Br-C*) and 2D (expression data for *vriIIe*). The new Figure 2A-D replaces Figure 2A-C of the original manuscript.

As we have shown before, RNAi depletion of EcR prevents significant gene activation whereas depletion of dMi-2 results in a super-activation. By contrast, gene activation measured in USP-depleted cells did not differ significantly from gene activation in GFP-depleted control cells throughout the 6 hour time course. These results provide further support for our hypothesis that hormonal regulation of *Br-C* and *vriIIe* expression depends on formation of an EcR/dMi-2 complex but does not require USP. The new data complement and strengthen our ChIP results (USP dispensable for dMi-2 recruitment to chromatin, Fig. 3C) and protein-protein interaction assays (EcR and dMi-2 form a complex that is devoid of USP, Figs 4A and 4B). We thank the reviewer for his/her suggestions.

- *Please indicate in the figure legend the treatment time and concentration of 20HE.*

We have added the requested information about treatment time and 20HE concentration to the figure legend.

- *Furthermore, the derepression experiments shown in Fig. 2D and 2E of *Br-Rb* and *vriIIe* as well as that of the two lncRNAs should include the RNAi experiments for USP.*

- *Authors should knockdown USP and analyze the expression of various other EcR responsive genes in the absence and presence of 20HE to generalise the conclusions.*

Figures 2D and 2E of the original manuscript reported expression levels of *Br-C*, *vriille* and two non-coding RNAs transcribed from the *vriille* locus in absence and presence of ecdysone in dMi-2, EcR and GFP (control) depleted cells. We agree with the reviewer that it is sensible to include USP depleted cells in the analysis and also to test more ecdysone-dependent genes in order to generalise our conclusions. Therefore, we have redone these RNAi / RT-qPCR experiments, including USP knockdown cells, and also analysed three additional dMi-2 bound genes (*Hr4*, *E23* and *let-7*) that we had confirmed to be activated by ecdysone (Fig. 1C). To consolidate and streamline the new data we have chosen to combine it into a single Figure 2E replacing Figures 2D and 2E of the original manuscript.

The new data confirm and extend the results reported in the original manuscript:

In the absence of hormone, we find widespread de-repression of basal transcription after dMi-2 knockdown (*Br-C*, *vriille*, *Hr4*, *E23*, *CR44742*, *CR44743*, *let-7*). The magnitude of de-repression obtained after dMi-2 knockdown ranged from 2.5-fold to 50-fold. By contrast, depletion of USP either failed to derepress transcription or led to a minor upregulation (max. 2-fold). EcR depletion deregulated the majority of the genes tested (1.5-fold to 20-fold). The finding that dMi-2 is required to restrict basal expression levels of all ecdysone-dependent genes tested here is in agreement with a general role of this remodeler in the repression of hormone-dependent genes. By contrast, repression of all genes tested was largely maintained in USP-depleted cells.

In the presence of hormone, dMi-2-depleted cells showed a general increase in transcription, whereas, as expected, depletion of EcR lowered the transcriptional output of all genes tested. By contrast, depletion of USP did not result in significant reductions of transcription.

Taken together, the combined data presented in the new Figure 2E, further strengthen the hypothesis that dMi-2 plays a general role in restricting the expression of hormone-regulated genes in both the absence and the presence of hormone. They confirm the role of EcR as an essential activator of ecdysone-induced transcription and show that it also functions to repress transcription in the absence of hormone. Our data does not support the view that USP plays a major role in regulating transcription of the ecdysone-dependent genes that we have tested.

4. Analyzing the data points of Fig. 6C and 6D there are rather sigmoidal and not a linearity of the data obtained with incubation of Mi-2 and EcR mediated remodelling activity as shown in Fig. 6C and Fig. 6D. What is the evidence to exclude a sigmoid interdependency? Authors should discuss the possibility of a sigmoid curve / interdependency.

The reviewer is correct in observing that the data points obtained in reactions combining dMi-2 and EcR could also be fitted to a sigmoidal curve.

We have used the one phase decay equation to fit our remodeling data as this is routinely used when data of this type is presented (see e.g. Bouazoune et al., PNAS, 109(47):19238-43 (2012)).

Sigmoidal graphs indicate interdependency/cooperativity effects when reaction velocity is plotted against substrate concentration (Michaelis-Menten). However, we are plotting product (% cut) vs. time. Here, lower apparent reaction rates at early time points (which give rise to a sigmoidal appearance) reflect the pre-steady state step of the reaction. This does not indicate interdependency/cooperativity between dMi-2 and EcR. This initial delay in the reaction is only detectable when reaction rates are high (i.e. in reactions where dMi-2 is stimulated by EcR).

Our data does not address interdependency/cooperativity between dMi-2 and EcR. Therefore, we choose not to speculate about this issue.

5. Please indicate the role of 20HE in Fig. 6 and show results in the presence of 20HE.

As requested by the reviewer we have now repeated the experiments demonstrating EcR-stimulation of dMi-2-mediated nucleosome remodeling in the presence of 20HE. In agreement with our finding that 20HE does not influence the interaction between dMi-2 and EcR in vitro (Figures

4A and 4C), we also do not detect an influence of 20HE on EcR-stimulation. We have added the new data as Figure 6A (bottom panel) and 6C.

6. The data suggest that EcR binds to the ATP binding site of dMi-2 and activates its remodelling activity. Please show whether this depends on ATP concentrations using the same experimental setup shown in Fig. 6D. Were saturating ATP concentrations used?

The reviewer raises a valid point: If stimulation of dMi-2-mediated nucleosome remodeling by EcR is only detected when ATP concentrations are limiting then this would indicate that EcR binding to the dMi-2 ATPase domain facilitates ATP binding, thereby providing a potential mechanism for stimulation. However, our nucleosome remodeling assays were performed with saturating ATP concentrations (2mM) which are routinely used in assays of this type. For the reviewers information we include here a figure demonstrating that ATP concentrations were indeed saturating. We have repeated the ATPase assay using one third of our standard ATP concentration. We observe no significant changes to basal dMi-2 and EcR-stimulated dMi-2 remodeling activity.

Thus, EcR stimulation is detected at high, saturating ATP concentrations suggesting that an increase in ATP affinity (decrease in K_m) is unlikely to have a significant impact on stimulation. We do not feel that this figure warrants inclusion in the manuscript.

7. Authors suggest that an EcR monomer is binding to dMi-2 on DNA. The affinity of a monomer of EcR might be too weak in vivo at chromatinized sites. Nuclear receptors are able to also dimerize with their DNA binding domain. What is the evidence that authors exclude a homodimer of EcR? If no clear proof of a monomeric EcR is provided authors should change their statements and the Fig. 7.

We thank the reviewer for this insightful comment. We agree that our data does not allow us to exclude homodimerisation of EcR in our model. As requested, we have changed the discussion (page 9/10; e.g. “We propose that ecdysone-regulated genes are not only occupied by EcR/USP heterodimers but, in addition, by EcR monomers and/or homodimers (that dimerise via their DNA binding domain)...”) and the legend to Figure 7.

Minor points:

8. Depict in Fig. 5A and 5B the linearity of both factors since the migration of the bands does not reflect the protein regions indicated in the schemes.

We are not sure what the reviewer means by “depict...the linearity of both factors”. We think the reviewer takes issue with the migration rate of the four EcR fragments tested not conforming to the fragment sizes depicted in the scheme. The four fragments analysed are AF1 (aa 1-227; length 227aa), DBD (aa 228-431; length 204 aa), AF2 (aa 432-655, length 224 aa) and CT (aa 656-878, length 223 aa). The MWs of these fragments are within 10% of each other. Therefore, one might expect them to show very similar migration behaviours during SDS PAGE (around 25 kDa). In fact,

the migration rates vary indicating apparent MWs ranging from 20 kDa to 35 kDa. This is not surprising. We often find that the actual migration rates of small polypeptides (10-30 kDa) deviate from the expected one. Migration rates during SDS-PAGE of small polypeptides is significantly influenced by amino acid composition (e.g. a high content of acidic amino acid residues increases the apparent MW). The four fragments analysed here show the same relative migration behaviour during SDS PAGE after expression using the baculovirus system (Figure 5) and after expression by *in vitro* translation (*data not shown*) indicating that migration behaviours are inherent to the fragments and do not dependent on the expression system. We hope this addresses the reviewer's concern.

9. Indicate in Fig. 1B legend the treatment time of 20HE.

The missing information has been added ("Cells were treated for 6 hours with 1uM 20HE.").

Reviewer #2

(reviewer comments in *italics*)

We thank the reviewer for his/her kind words and for their assessment that our findings add much detail to the current bimodal switch model. We are pleased, that the reviewer finds that our study makes an important advance in our understanding of transcriptional regulation by nuclear hormone receptors.

Reviewer #3

(reviewer comments in *italics*)

We are delighted that the reviewer finds that our study provides "important new insight in comparison to previous studies suggesting that the role of dMi-2 was to maintain the inactive state".

One of the most interesting aspects of the manuscripts is that it identifies contacts between EcR and the ATPase domain of dMi-2. Effects on the activity of dMi-2 are shown using a restriction enzyme accessibility assay using a 230 bp DNA fragment. It's possible that EcR could bind this DNA non-specifically and affect the outcome of ATP-dependent reactions. As a result I don't think it is safe to conclude that EcR stimulates activity from this assay. It would be compelling if the authors could show an effect on ATPase activity as this would be independent of any effect EcR has on the template.

The reviewer makes an important point. Indeed, we cannot fully exclude that binding of EcR to the nucleosome as opposed to binding of EcR to the dMi-2 ATPase domain is contributing to stimulation of dMi-2 remodeling activity (indeed, EcR does bind the nucleosome under assay conditions, see below). In essence, in this scenario an EcR-bound nucleosome would constitute a much better substrate for dMi-2-mediated remodeling than an unbound nucleosome. We would like to note that this might very well contribute to guiding dMi-2 remodeling activity to specific nucleosomes (with EcR binding sites in the vicinity) *in vivo* and does not argue against our main hypothesis that EcR-stimulates dMi-2 mediated remodeling. To account for this possibility we have included the following section in the discussion: "The physical interaction between EcR and dMi-2 described in this study goes beyond recruiting the remodeler to chromatin: EcR also stimulates dMi-2 nucleosome remodeling activity *in vitro*. It is possible that an EcR-bound nucleosome provides a better substrate for dMi-2-mediated remodeling than the nucleosome alone. However, we favour the hypothesis that the stimulation of dMi-2 nucleosome remodeling activity results from EcR contacting the ATPase domain of dMi-2."

The reviewer suggests to distinguish between the two mechanisms, namely to test if EcR stimulates ATPase activity of dMi-2 as this would be "independent of any effect EcR has on template". However, dMi-2 is a nucleosome-stimulated ATPase and significant dMi-2 ATPase

activity can only be detected in the presence of nucleosomes (e.g. Brehm A. *et al*, EMBO J., **15**, 4332-4341 (2000)). So, nucleosomes are also present in the ATPase reaction and, therefore, any stimulation of dMi-2 ATPase activity observed after addition of EcR could again be accounted for both by binding of EcR to the ATPase domain as well as by binding of EcR to the nucleosome substrate.

Nevertheless, we have followed the reviewer's suggestion and performed dMi-2 ATPase assays in the presence of EcR. We have performed these assays multiple times with different preparations of dMi-2, EcR and nucleosomes. A typical experiment that illustrates the difficulties in interpreting these results is shown below. We have found that EcR copurifies with significant phosphatase activity when we prepare recombinant EcR. This EcR-associated phosphatase activity is not stimulated by nucleosomes and EcR does not remodel nucleosomes in the absence of added dMi-2 (Figure 6) demonstrating that the copurifying phosphatase activity does not interfere with our nucleosome remodeling assays. However, with this "background" phosphatase activity it is difficult to detect stimulation of dMi-2 ATPase function by EcR. We do see that combination of dMi-2 and EcR produces ATPase activity levels (dMi-2+EcR+nuc: 21.2% hydrolysis) that are greater than the sum of ATPase activities associated with the individual proteins (EcR+nuc: 7.1%, dMi-2+nuc: 6.6%), suggesting that, indeed, EcR might stimulate dMi-2 ATPase activity. However, given the presence of two ATP hydrolysing activities in the assay (dMi-2 and EcR-associated) we do not feel confident about drawing any strong conclusions and prefer to not include this data in the manuscript.

An alternative explanation is that the interaction between EcR and the ATPase domain acts to recruit dMi-2 rather than increase its turnover. This would be a more conventional mode of action and in this case experiments could be performed using templates with and without EcR binding sites.

As mentioned above, EcR does bind the mononucleosome under conditions of the remodeling assay even though it lacks *bona fide* EcR binding sites (*data not shown*), so performing the assay with a nucleosome substrate containing EcR binding sites is unlikely to change the result. It is important to note that in this type of remodeling reaction, the remodeler (and, in this case, also EcR) is present in excess over the mononucleosome substrate. As a consequence, binding of dMi-2 to the nucleosome is not limiting and nucleosomes are quantitatively bound by dMi-2 when assayed by bandshift (*data not shown*) even in the absence of EcR. Nucleosome binding by dMi-2 and EcR are driven by mass action and not the result of sequence-specific DNA binding. Given that dMi-2 binding to the nucleosome substrate is not limiting under our assay conditions we consider a recruitment mechanism unlikely.

Changes to chromatin are assessed at dMi-2 dependent genes by MNase digestion and PCR. It would be much more powerful to use MNase-seq as this would provide information about changes to nucleosome positioning at all EcR loci, rather than the evidence for generic changes to accessibility provided by the current data.

While we completely agree with the reviewer that getting a genome-wide view of changes to nucleosome positioning at all EcR binding sites we feel that this would be beyond the scope of our manuscript which mostly focuses on biochemical findings (identification of a novel EcR/dMi-2 complex, competitive binding of dMi-2 and USP to EcR, EcR contacting the dMi-2 ATPase domain, stimulation of dMi-2-mediated remodeling in vitro). Our results that dMi-2 regulates chromatin accessibility at the vrille locus provides a proof-of-principle for a potential mechanism of action.

REVIEWERS' COMMENTS:

Reviewer #1 (Remarks to the Author):

Authors have addressed all criticism in a full satisfactory manner.

Reviewer #3 (Remarks to the Author):

From the comments in the rebuttal it is clear that the authors believe that EcR most likely promotes recruitment of dMi-2 to nucleosomes rather than increasing its catalytic rate of turnover. This is not clear to me from the first version of the manuscript and I think careful thought needs to be given to phrases such as "EcR stimulates dMi-2 nucleosome remodeling activity *in vitro*". Strictly activity can be considered as combined effects of K_m and K_{cat} , so the above statement is correct, however, I interpreted the original version of the manuscript as indicating an effect on K_{cat} . It's now clear this is not likely to be the case and as a result the impact of the work is reduced, but the study still provides new insights.

The proteins used in the assays are not pure enough for kinetic parameters to be calculated. However, it's clear from the rebuttal that the authors know that EcR binding to nucleosomes lacking EcR sites under the conditions of the assay. Also EcR alone does not stimulate ATPase activity. This should be stated clearly in the manuscript to favor a mechanism acting via recruitment.

The authors should carefully consider whether the effects on activity are being used in a misleading way throughout the manuscript including in the title. For example, the following statement needs to be reworded. "The physical interaction between EcR and dMi-2 described in this study goes beyond recruiting the remodeler to chromatin:..." The most likely explanation is that EcR does stimulate simply by recruitment. It's just that the recruitment is via the ATPase domain.

The last two sentences of the abstract "Unexpectedly, EcR contacts the 47 dMi-2 ATPase domain and stimulates its remodeling activity. This study identifies a novel non-48 canonical EcR-corepressor complex and exposes the first direct regulation of ATP-dependent 49 remodeling activity by a nuclear hormone receptor. " can be interpreted as indicating that contacts with EcR increase the turnover rate of the enzyme and need to be re-written.

minor:

This section needs revision as it refers to yeast Chd1 and the chromodomains of yeast Chd1 are unlikely to recognize H3 K4 Me.

The remodeling activity of Chd1 is inhibited by an intramolecular interaction between its chromodomains and its ATPase domain 44 368 - 369 Chromodomain binding to nucleosomes carrying H3K4 methylation disrupts this inhibitory 370 interaction and stimulates ATPase and remodeling activity.

Response to reviewer #3:

We thank the reviewer for taking time again to evaluate our manuscript.

The reviewer has one remaining major issue and one minor point.

Major issue:

“From the comments in the rebuttal it is clear that the authors believe that EcR most likely promotes recruitment of dMi-2 to nucleosomes rather than increasing its catalytic rate of turnover. This is not clear to me from the first version of the manuscript and I think careful thought needs to be given to phrases such as “EcR stimulates dMi-2 nucleosome remodeling activity in vitro”. Strictly activity can be considered as combined effects of K_m and K_{cat} , so the above statement is correct, however, I interpreted the original version of the manuscript as indicating an effect on K_{cat} . It’s now clear this is not likely to be the case and as a result the impact of the work is reduced, but the study still provides new insights.

The proteins used in the assays are not pure enough for kinetic parameters to be calculated. However, it's clear from the rebuttal that the authors know that EcR is binding to nucleosomes lacking EcR sites under the conditions of the assay. Also EcR alone does not stimulate ATPase activity. This should be stated clearly in the manuscript to favor a mechanism acting via recruitment.

The authors should carefully consider whether the effects on activity are being used in a misleading way throughout the manuscript including in the title. For example, the following statement needs to be reworded. “The physical interaction between EcR and dMi-2 described in this study goes beyond recruiting the remodeler to chromatin:...” The most likely explanation is that EcR does stimulate simply by recruitment. It's just that the recruitment is via the ATPase domain.

The last two sentences of the abstract "Unexpectedly, EcR contacts the 47 dMi-2 ATPase domain and stimulates its remodeling activity. This study identifies a novel non-48 canonical EcR-corepressor complex and exposes the first direct regulation of ATP-49 dependent remodeling activity by a nuclear hormone receptor. " can be interpreted as indicating that contacts with EcR increase the turnover rate of the enzyme and need to be re-written."

We appreciate the remaining concern of reviewer #3 although we have a different opinion about the relative likelihood of the different molecular mechanisms that potentially underlie our observations.

The issue here is whether the strong increase in dMi-2-mediated nucleosome remodelling in vitro observed in the presence of EcR is a consequence of (a) EcR contacting the dMi-2 ATPase domain and in doing so modify kinetic parameters of the enzyme or is a result of (b) EcR simply recruiting dMi-2 (via binding the ATPase domain) to the nucleosome substrate.

The reviewer has come to the conclusion that the observed effects are due to recruitment of dMi-2 to the mononucleosome by EcR. She/he states that even “the authors believe that EcR most likely promotes recruitment of dMi-2 to nucleosomes rather than increasing its catalytic rate of turnover.” We disagree with this assessment. In fact, we have tried to make it very clear that we consider recruitment to be very unlikely to play a significant role in increasing dMi-2-mediated nucleosome remodelling in vitro.

As we have explained in the response to the reviewers, under the conditions of our Restriction Enzyme Accessibility (REA) assay binding of dMi-2 to nucleosomes is not limiting. In a bandshift assay carried out with the same dMi-2 and nucleosome concentrations used in the REA assay all nucleosomes are bound by dMi-2 (in the absence of EcR). This is a consequence of mass action and the strong sequence-nonspecific DNA and nucleosome-binding activity of dMi-2 that is well established (e.g. Brehm et al, EMBO J, 2000; Bouazoune et al, EMBO J, 2002). In other words, in this in vitro assay dMi-2 does not need the help of a transcription factor to bind to its substrate. Therefore, we consider it very unlikely that recruitment by EcR makes a substantial contribution to the observed stimulation of dMi-2 remodelling activity in vitro and we favour a model that involves conformational changes increasing the enzymatic activity of dMi-2, similar to conformational changes reported recently for other remodelers (Chd1, ISWI).

Of course, we cannot formally exclude that recruitment makes a contribution and we cannot quantify the relative contributions of recruitment, stimulation of dMi-2 remodelling activity or other mechanisms (e.g. EcR-bound nucleosomes being better substrates for dMi-2).

The reviewer rightly criticises the use of statements such as “EcR stimulates dMi-2 nucleosome remodelling activity in vitro”. He feels that such statements imply that EcR by contacting the dMi-2 ATPase domain changes *k_{cat}*. We agree with the reviewer that this phrasing is too strong and can be misleading.

Since it is impossible at this point to provide conclusive evidence that EcR binding to the dMi-2 ATPase domain increases the catalytic parameters of the enzyme and that this mechanism makes a larger contribution than other potential mechanisms we have rephrased our statements.

Essentially, we have changed all statements that “EcR stimulates dMi-2 nucleosome remodelling activity in vitro” to “EcR increases the efficiency of a dMi-2-mediated nucleosome remodelling reaction in vitro”. This wording focuses on the outcome of the remodelling reaction (which can be influenced by different mechanisms) rather than on the enzyme catalysing the reaction.

We have changed the following sections of the manuscript to accommodate the reviewer's concern:

Title

“EcR recruits and stimulates dMi-2 remodelling activity to constrain transcription of hormone-regulated genes” was changed to “EcR recruits dMi-2 and increases the efficiency of dMi-2-mediated remodelling to constrain transcription of hormone-regulated genes”

Abstract

“Unexpectedly, EcR contacts the dMi-2 ATPase domain and stimulates its remodelling activity. This study identifies a novel non-canonical EcR-corepressor complex and exposes the first direct regulation of ATP-dependent remodelling activity by a nuclear hormone receptor.” was changed to “Unexpectedly, EcR contacts the dMi-2 ATPase domain and increases the efficiency of dMi-2 mediated nucleosome remodelling. This study identifies a novel non-canonical EcR-corepressor complex with the potential for a direct regulation of ATP-dependent nucleosome remodeling by a nuclear hormone receptor.”

Main text

“*EcR stimulates dMi-2 nucleosome remodelling activity in vitro*” was changed to “EcR increases dMi-2-mediated nucleosome remodelling *in vitro*” (p.8)

“We conclude that EcR, but not USP, stimulates the nucleosome remodelling activity of dMi-2 *in vitro*.” was changed to “We conclude that EcR, but not USP, increases dMi-2-mediated nucleosome remodelling *in vitro*.” (p.9)

“The physical interaction between EcR and dMi-2 described in this study goes beyond recruiting the remodeler to chromatin: EcR also stimulates dMi-2 nucleosome remodelling activity *in vitro*.” was changed to “The physical interaction between EcR and dMi-2 described in this study potentially goes beyond recruiting the remodeler to chromatin.” (p.11)

Minor point:

“This section needs revision as it refers to yeast Chd1 and the chromodomains of yeast Chd1 are unlikely to recognized H3 K4 Me.

The remodeling activity of Chd1 is inhibited by an intramolecular interaction between its chromodomains and its ATPase domain 44 368 . 369 Chromodomain binding to nucleosomes carrying H3K4 methylation disrupts this inhibitory 370 interaction and stimulates ATPase and remodeling activity.”

The reviewer correctly points out that yeast Chd1 is unlikely to recognise K4 methylated nucleosomes. We, therefore, have changed “Chromodomain binding to nucleosomes carrying H3K4 methylation disrupts this inhibitory interaction and stimulates ATPase and remodelling activity.” to “Chromodomain binding to nucleosomes disrupts this inhibitory interaction and stimulates ATPase and remodelling activity.”